# POSTERIOR SAMPLING BASED ON GRADIENT FLOWS OF THE MMD WITH NEGATIVE DISTANCE KERNEL

**P. Hagemann**[1], **J. Hertrich**[2], **F. Altekrüger**[3], **R. Beinert**[1], **J. Chemseddine**[1], **G. Steidl**[1]
[1] Technische Universität Berlin, [2] University College London, [3] Humboldt-Universität zu Berlin
Correspondence to: `hagemann@math.tu-berlin.de`

## ABSTRACT

We propose conditional flows of the maximum mean discrepancy (MMD) with the negative distance kernel for posterior sampling and conditional generative modelling. This MMD, which is also known as energy distance, has several advantageous properties like efficient computation via slicing and sorting. We approximate the joint distribution of the ground truth and the observations using discrete Wasserstein gradient flows and establish an error bound for the posterior distributions. Further, we prove that our particle flow is indeed a Wasserstein gradient flow of an appropriate functional. The power of our method is demonstrated by numerical examples including conditional image generation and inverse problems like superresolution, inpainting and computed tomography in low-dose and limited-angle settings.

## 1 INTRODUCTION

The tremendous success of generative models led to a rising interest in their application for inverse problems in imaging. Here, an unknown image $x$ has to be recovered from a noisy observation $y = f(x) + \xi$. Since the forward operator $f$ is usually ill-conditioned, such reconstructions include uncertainties and are usually not unique. As a remedy, we take a Bayesian viewpoint and consider $x$ and $y$ as samples from random variables $X$ and $Y$, and assume that we are given training samples from their joint distribution $P_{X,Y}$. In order to represent the uncertainties in the reconstruction, we aim to find a process to sample from the posterior distributions $P_{X|Y=y}$. This allows not only to derive different possible predictions, but also to consider pixel-wise standard deviations for identifying highly vague image regions. Figure 1 visualizes this procedure on an example for limited angle computed tomography.

Nowadays, generative models like (Wasserstein) GANs (Arjovsky et al., 2017; Goodfellow et al., 2014) and VAEs (Kingma & Welling, 2014) have turned out to be a suitable tool for approximating probability distributions. In this context, the field of gradient flows in measure spaces received increasing attention. Welling & Teh (2011) proposed to apply the Langevin dynamics in order to generate samples from a known potential, which corresponds to simulating a Wasserstein gradient flow with respect to the Kullback-Leibler (KL) divergence, see Jordan et al. (1998). Score-based and diffusion models extend this approach by estimating the gradients of the potential from training data, see (De Bortoli et al., 2021; Ho et al., 2020; Song & Ermon, 2020; Song et al., 2021) and achieved state-of-the-art results. The simulation of Wasserstein gradient flows with other functionals than KL, based on the JKO scheme, was considered in Altekrüger et al. (2023c); Alvarez-Melis et al. (2022); Fan et al. (2022); Mokrov et al. (2021).

In this paper, we focus on gradient flows with respect to MMD with negative distance kernel $K(x,y) = -\|x-y\|$, which is also known as energy distance, see (Sejdinovic et al., 2013; Székely, 2002; Székely & Rizzo, 2009; 2013). While MMDs have shown great success at comparing two distributions in general, see (Gretton et al., 2012; Székely & Rizzo, 2005; Gretton et al., 2006), their combination with the negative

distance kernel results in many additional desirable properties as translation and scale equivariance (Székely & Rizzo, 2013), efficient computation (Hertrich et al., 2024), a dimension independent sample complexity of $O(n^{-1/2})$ (Gretton et al., 2012) and unbiased sample gradients Bellemare et al. (2017).

To work with probability distributions in high dimensions, Rabin et al. (2012) proposed to slice them. Applied on gradient flows, this leads to a significant speed-up, see (Du et al., 2023; Kolouri et al., 2019b; Liutkus et al., 2019). In particular, for MMD with negative distance kernel slicing does not change the metric itself and reduces the time complexity of calculating gradients from $O(N^2)$ to $O(N \log N)$ for measures with $N$ support points, see Hertrich et al. (2024).

In order to use generative models for inverse problems, an additional conditioning parameter was added to the generation process in (Ardizzone et al., 2019; 2021; Chung et al., 2023; Hagemann et al., 2022; Mirza & Osindero, 2014). However, this approach cannot directly applied to the gradient flow setting, where the generator is not trained end-to-end.

**Contributions.** We simulate conditional MMD particle flows for posterior sampling in Bayesian inverse problems. To this end, we provide three kinds of contributions. The first two address theoretical questions while the last one validates our findings numerically.

- Conditional generative models approximate the joint distribution by learning a mapping $T$ such that $P_{X,Y} \approx P_{T(Z,Y)),Y}$, but in fact we are interested in the posterior distributions $P_{X|Y=y}$. In this paper, we prove error bounds between posterior and joint distributions within the MMD metric in expectation. The proofs of these results are based on relations between measure spaces and RKHS as well as Lipschitz stability results under pushforwards.

- We represent the considered particle flows as Wasserstein gradient flows of a modified MMD functional. As a side effect of this representation, we can provide a theoretical justification for the empirical method presented by Du et al. (2023), where the authors obtain convincing results by neglecting the velocity in the $y$-component in sliced Wasserstein gradient flows. Based on locally isometric embeddings of the $\mathbb{R}^{Nd}$ into the Wasserstein space, we can show that the result is again a Wasserstein gradient flow with respect to a modified functional.

- We approximate our particle flows by conditional generative neural networks and apply the arising generative model in various settings. On the one hand, this includes standard test sets like conditional image generation and inpainting on MNIST, FashionMNIST and CIFAR10 and superresolution on CelebA. On the other hand, we consider very high-dimensional imaging inverse problems like superresolution of materials' microstructures as well as limited-angle and low-dose computed tomography.

**Related work.** Many generative models like GANs, VAEs and normalizing flows can be used for posterior sampling by adding a conditioning parameter as an additional input, see (Ardizzone et al., 2019; 2021; Batzolis et al., 2021; Hagemann et al., 2022; Mirza & Osindero, 2014). The loss function of these methods is based on the Kullback–Leibler divergence. In this case, stability results were proven by Altekrüger et al. (2023b) based on local Lipschitz regularity results from Sprungk (2020).

In this paper, we are interested in generative models which are based on gradient flows (Fan et al., 2022; Kolouri et al., 2019b; Hertrich et al., 2024; 2023b; Nguyen & Ho, 2022; Mokrov et al., 2021). In this case, the above approach is not directly applicable since the network is not trained end-to-end. For the sliced Wasserstein gradient flows, Du et al. (2023) proposed to approximate the joint distribution while neglecting the velocity in one component. They achieved very promising results, but evaluated their model empirically without giving theoretical justification.

Here, we consider gradient flows with respect to MMD with negative distance kernel, which is also known as energy distance or Cramer distance (Sejdinovic et al., 2013; Székely, 2002). The theoretical analysis

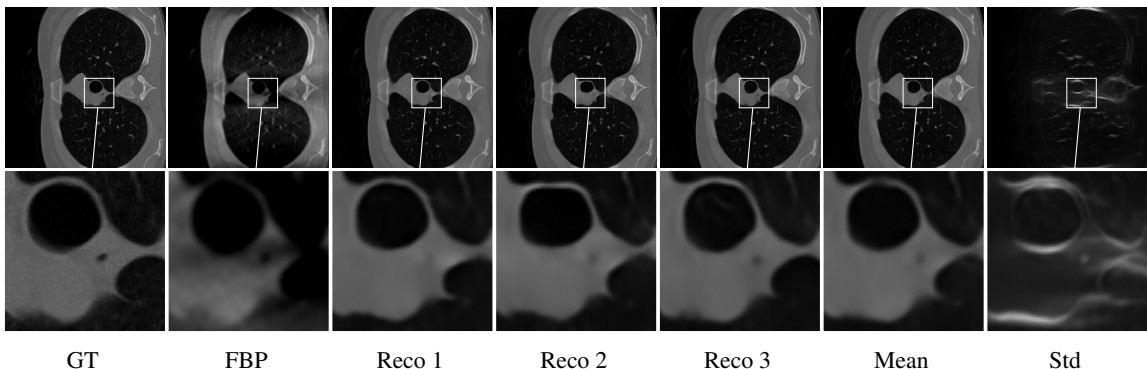

| GT | FBP | Reco 1 | Reco 2 | Reco 3 | Mean | Std |

Figure 1: Generated posterior samples, mean image and pixel-wise standard deviation for limited angle computed tomography using conditional MMD flows.

of such gradient flows in the Wasserstein space turns out to be challenging, since discrete measures might become absolutely continuous and vice versa, see (Hertrich et al., 2023b). As a remedy, many papers consider particle flows as a space discretization, see (Carrillo et al., 2020; Daneshmand & Bach, 2023; Daneshmand et al., 2023; Hertrich et al., 2024). The question whether the mean-field limit of these particle flows corresponds to the continuous Wasserstein gradient flow is so far only partially answered and still an active area of research, see (Carrillo et al., 2020; Daneshmand & Bach, 2023; Daneshmand et al., 2023). Further, there is plenty of literature covering the statistical properties of MMD in general (Sriperumbudur et al., 2011; Gretton et al., 2012; Modeste & Dombry, 2023) as well as its applications to causal inference (Kremer et al., 2022; 2023) via conditional moments.

**Outline of the paper.** Section 2 briefly recalls MMD with negative distance kernel and corresponding discrete Wasserstein gradient flows. Based on this, we introduce conditional generative MMD flows in Section 3 by the following path: i) we establish relations between joint and posterior distributions, ii) we present an interpretation of the conditional particle flow as Wasserstein gradient flow of an appropriate functional, and iii) we suggest a generative variant of the conditional MMD flow. Numerical experiments are contained in Section 4 and conclusions are drawn in Section 5. The appendix contains all the proofs, implementation details and additional experimental results.

## 2 MMD AND DISCRETE GRADIENT FLOWS

Let $\mathcal{P}_p(\mathbb{R}^d)$, $p \in (0, \infty)$ denote the space of probability measures with finite $p$-th moments. We are interested in the MMD $\mathcal{D}_K \colon \mathcal{P}_2(\mathbb{R}^d) \times \mathcal{P}_2(\mathbb{R}^d) \to \mathbb{R}$ of the negative distance kernel $K(x, y) \coloneqq -\|x - y\|$ defined by

$$
\begin{aligned}
\mathcal{D}_K^2(\mu, \nu) \coloneqq{} & \frac{1}{2} \int_{\mathbb{R}^d} \int_{\mathbb{R}^d} K(x, y) \, \mathrm{d}\mu(x) \, \mathrm{d}\mu(y) - \int_{\mathbb{R}^d} \int_{\mathbb{R}^d} K(x, y) \, \mathrm{d}\mu(x) \, \mathrm{d}\nu(y) \\
& + \frac{1}{2} \int_{\mathbb{R}^d} \int_{\mathbb{R}^d} K(x, y) \, \mathrm{d}\nu(x) \, \mathrm{d}\nu(y).
\end{aligned}
\tag{1}
$$

It is a metric on $\mathcal{P}_1(\mathbb{R}^d) \supset \mathcal{P}_2(\mathbb{R}^d)$, see e.g. Sejdinovic et al. (2013); Székely & Rizzo (2013). In particular, we have that $\mathcal{D}_K(\mu, \nu) = 0$ if and only if $\mu = \nu$.

For a fixed measure $\nu \in \mathcal{P}_2(\mathbb{R}^d)$, we consider Wasserstein gradient flows of the functional $\mathcal{F}_\nu \colon \mathcal{P}_2(\mathbb{R}^d) \to (-\infty, \infty]$ defined by

$$
\mathcal{F}_\nu(\mu) \coloneqq \mathcal{D}_K^2(\mu, \nu) - \mathrm{const},
\tag{2}
$$

where the constant is just the third summand in (1). For a definition of Wasserstein gradient flows, we refer to Appendix A.3. Since the minumum of this functional is $\nu$, we can use the flows to sample from this target distribution. While the general analysis of these flows is theoretically challenging, in particular for the above non-smooth and non-$\lambda$-convex negative distance kernel, see Carrillo et al. (2020); Hertrich et al. (2023b), we focus on the efficient numerical simulation via particle flows as proposed in Hertrich et al. (2024). To this end, let $P_N := \{\frac{1}{N}\sum_{i=1}^N \delta_{x_i} : x_i \in \mathbb{R}^d, x_i \neq x_j, i \neq j\}$ denote the set of empirical measures on $\mathbb{R}^d$ with $N$ pairwise different anchor points. Given $M$ independent samples $p = (p_1, ..., p_M) \in (\mathbb{R}^d)^M$ of the measure $\nu$, we deal with its empirical version $\nu_M := \frac{1}{M}\sum_{i=1}^M \delta_{p_i}$ and consider the Euclidean gradient flow of the *discrete MMD functional* $F_p\colon (\mathbb{R}^d)^N \to \mathbb{R}$ defined for $x = (x_1, ..., x_N) \in (\mathbb{R}^d)^N$ by

$$F_p(x) := \mathcal{F}_{\nu_M}(\mu_N) = -\frac{1}{2N^2}\sum_{i=1}^N\sum_{j=1}^N \|x_i - x_j\| + \frac{1}{MN}\sum_{i=1}^N\sum_{j=1}^M \|x_i - p_j\|. \tag{3}$$

Then, a curve $u = (u_1, ..., u_N)\colon [0, \infty) \to (\mathbb{R}^d)^N$ solves the ODE

$$\dot{u} = -N\nabla F_p(u), \quad u(0) = (u_1^{(0)}, ..., u_N^{(0)}),$$

if and only if the curve $\gamma_N\colon (0, \infty) \to \mathcal{P}_2(\mathbb{R}^d)$ defined by $\gamma_N(t) = \frac{1}{N}\sum_{i=1}^N \delta_{u_i(t)}$ is a Wasserstein gradient flows with respect to the functional $\mathcal{J}_{\nu_M}\colon \mathcal{P}_2(\mathbb{R}^d) \to \mathbb{R} \cup \{\infty\}$ given by

$$\mathcal{J}_{\nu_M}(\mu) := \begin{cases} \mathcal{F}_{\nu_M}(\mu), & \text{if } \mu \in P_N, \\ \infty, & \text{otherwise.} \end{cases}$$

In Hertrich et al. (2024), this simulation of MMD flows was used to derive a generative model.

## 3 CONDITIONAL MMD FLOWS FOR POSTERIOR SAMPLING

In this section, we propose *conditional* flows for posterior sampling. We consider two random variables $X \in \mathbb{R}^d$ and $Y \in \mathbb{R}^n$. Then, we aim to sample from the posterior distribution $P_{X|Y=y}$. One of the most important applications for this are Bayesian inverse problems. Here $X$ and $Y$ are related by $Y = \mathrm{noisy}(f(X))$, where $f\colon \mathbb{R}^d \to \mathbb{R}^n$ is some ill-posed forward operator and "noisy" denotes some noise process. Throughout this paper, we assume that we are only given samples from the joint distribution $P_{X,Y}$. In order to sample from the posterior distribution $P_{X|Y=y}$, we will use conditional generative models. More precisely, we aim to find a mapping $T\colon \mathbb{R}^d \times \mathbb{R}^n \to \mathbb{R}^d$ such that

$$T(\cdot, y)_{\#}P_Z = P_{X|Y=y}, \tag{4}$$

where $P_Z$ is an easy-to-sample latent distribution and $T(\cdot, y)_{\#}P_Z = P_Z(T^{-1}(\cdot, y))$ defines the pushforward of a measure. The following proposition summarizes the main principle of conditional generative modelling and provides a sufficient criterion for (4). To make the paper self-contained, we add the proof in Appendix A.1.

**Proposition 1.** *Let $X, Z \in \mathbb{R}^d$ be independent random variables and $Y \in \mathbb{R}^n$ be another random variable. Assume that $T\colon \mathbb{R}^d \times \mathbb{R}^n \to \mathbb{R}^d$ fulfills $P_{T(Y,Z),Y} = P_{X,Y}$. Then, it holds $P_Y$-almost surely that $T(\cdot, y)_{\#}P_Z = P_{X|Y=y}$.*

In Subsection 3.1, we extend this result and show that (4) still holds true approximately, whenever the distributions $P_{T(Y,Z),Y}$ and $P_{X,Y}$ are close to each other. Then, in Subsection 3.2, we construct such a mapping $T$ based on Wasserstein gradient flows with respect to a conditioned version of the functional $\mathcal{F}_{P_{X,Y}}$. Finally, we propose an approximation of this Wasserstein gradient flow by generative neural networks in Subsection 3.3.

## 3.1 POSTERIOR VERSUS LEARNED JOINT DISTRIBUTION

In Proposition 1, we assume that $T$ is perfectly learned, which is rarely the case in practice. Usually, we can only approximate the joint distribution such that $\mathcal{D}_K(P_{T(Z,Y),Y}, P_{X,Y})$ becomes small. Fortunately, we can prove under moderate additional assumptions that then the expectation with respect to $y$ of the distance of the posteriors $\mathcal{D}_K\left(T(\cdot,y)_{\#}P_Z, P_{X|Y=y}\right)$ becomes small too. A similar statement was shown by Kim et al. (2023, Proposition 3); note that their RHS is not equal to the MMD of joint distribution, but a modified version. Such a statement involving MMDs can not generally hold true, see Example 6.

**Theorem 2.** *Let $S_n \subset \mathbb{R}^n$ and $S_d \subset \mathbb{R}^d$ be compact sets. Further, let $X, \tilde{X} \in S_d$ and $Y \in S_n$ be absolutely continuous random variables, and assume that $P_{X,Y}$ and $P_{\tilde{X},Y}$ have densities fulfilling*

$$|p_{X|Y=y_1} - p_{X|Y=y_2}| \leq C_{S_n} \|y_1 - y_2\|^{\frac{1}{2}}, \tag{5}$$

$$|p_{\tilde{X}|Y=y_1} - p_{\tilde{X}|Y=y_2}| \leq C_{S_n} \|y_1 - y_2\|^{\frac{1}{2}} \tag{6}$$

*a.e. on $S_d$ for all $y_1, y_2 \in S_n$. Then it holds*

$$\mathbb{E}_{y \sim P_Y}[\mathcal{D}_K(P_{\tilde{X}|Y=y}, P_{X|Y=y})] \leq C\, \mathcal{D}_K(P_{\tilde{X},Y}, P_{X,Y})^{\frac{1}{4(d+n+1)}}. \tag{7}$$

The proof, which uses relations between measure spaces and reproducing kernel Hilbert spaces (RKHS), is given in Appendix A.2. Similar relations hold true for other „distances" as the Kullback–Leibler divergence or the Wasserstein distance, see Remark 7. For the latter one as well as for the MMD, it is necessary that the random variables are compactly supported. Example 6 shows that (7) is in general not correct if this assumption is neglected.

Based on Theorem 2, we can prove pointwise convergence of a sequence of mappings in a similar way as it was done for Wasserstein distances by Altekrüger et al. (2023b). For this, we require the existence of $C, \tilde{C} > 0$ such that

$$\mathcal{D}_K(T(\cdot,y_1)_{\#}P_Z, T(\cdot,y_2)_{\#}P_Z) \leq C\|y_1 - y_2\|^{\frac{1}{2}} \quad \text{(stability under pushforwards)}, \tag{8}$$

$$\mathcal{D}_K(P_{X|Y=y_1}, P_{X|Y=y_2}) \leq \tilde{C}\|y_1 - y_2\|^{\frac{1}{2}} \quad \text{(stability of posteriors)} \tag{9}$$

for all $y_1, y_2 \in S_n$ with $p_Y(y_1), p_Y(y_2) > 0$. In Appendix A.2, we show that both stability estimates hold true under certain conditions, see Lemma 9 and 10. Furthermore, we prove following theorem.

**Theorem 3.** *Let $S_n \subset \mathbb{R}^n$ and $S_d \subset \mathbb{R}^d$ be compact sets and $X, Z \in S_d$ and $Y \in S_n$ absolutely continuous random variables. Assume that $|p_Y(y_1) - p_Y(y_2)| \leq C'\|y_1 - y_2\|^{\frac{1}{2}}$ for all $y_1, y_2 \in S_n$ and fixed $C' > 0$. Let $P_{X,Y}$ have a density satisfying (5). Moreover, let $\{T^{\varepsilon} : \mathbb{R}^d \times \mathbb{R}^n \to \mathbb{R}^d\}$ be a family of measurable mappings with $\mathcal{D}_K(P_{T^{\varepsilon}(Z,Y),Y}, P_{X,Y}) \leq \varepsilon$, which fulfill (8) and (9) with uniform $C, \tilde{C} > 0$. Further, assume that $P_{\tilde{X},Y}$ with $\tilde{X} = T^{\varepsilon}(Z,Y)$ have densities satisfying (6) with uniform $C_{S_n} > 0$. Then, for all $y \in S_n$ with $p_Y(y) > 0$, it holds*

$$\mathcal{D}_K(T^{\varepsilon}(\cdot,y)_{\#}P_Z, P_{X|Y=y}) \to 0 \quad as \quad \varepsilon \to 0.$$

## 3.2 CONDITIONAL MMD FLOWS

In the following, we consider particle flows to approximate the joint distribution $P_{X,Y}$. Together with the results from the previous subsection, this imposes an approximation of the posterior distributions $P_{X|Y=y}$. Let $N$ pairwise distinct samples $(p_i, q_i) \in \mathbb{R}^d \times \mathbb{R}^n$ from the joint distribution $P_{X,Y}$ be given, and set $(p,q) \coloneqq ((p_i, q_i))_{i=1}^N$. Let $P_Z$ be a $d$-dimensional latent distribution, where we can easily sample from. We draw a sample $z_i$ for each $i$ and consider the particle flow $t \mapsto (u(t), q)$ starting at $((z_i, q_i))_{i=1}^N$, where the

second component remains fixed and the first component $u = (u_1, ..., u_N) \colon [0, \infty) \to \mathbb{R}^{dN}$ follows the gradient flow

$$\dot{u}(t) = -N \nabla_x F_{(p,q)}\left((u, q)\right), \quad u(0) = (z_1, \dots, z_N), \tag{10}$$

with the function $F_{(p,q)}$ in (3) and the gradient $\nabla_x$ with respect to the first component. Since the MMD is a metric, the function $x \mapsto F_{(p,q)}((x, q))$ admits the global minimizer $x = (p_1, ..., p_N)$. In our numerical examples, we observe that the gradient flow (10) approaches this global minimizer as $t \to \infty$. Finally, we approximate the motion of the particles by a mapping $T \colon \mathbb{R}^d \times \mathbb{R}^n \to \mathbb{R}^d$, which describes how the initial particle $(z_i, q_i)$ moves to the particle $u_i(t_{\max})$ at some fixed time $t_{\max}$. Moreover, by the convergence of the gradient flow (10), we have that

$$P_{T(Z,Y),Y} \approx \frac{1}{N} \sum_{i=1}^{N} \delta_{T(z_i,q_i),q_i} = \frac{1}{N} \sum_{i=1}^{N} \delta_{u_i(t_{\max}),q_i} \approx \frac{1}{N} \sum_{i=1}^{N} \delta_{p_i,q_i} \approx P_{X,Y}.$$

In particular, the arising mapping $T$ fulfills the assumptions from the previous subsection such that we obtain

$$T(\cdot, y)_{\#} P_Z \approx P_{X|Y=y}, \quad \text{for } P_Y - \text{a.e. } y \in \mathbb{R}^n.$$

The following theorem states that the solutions of (10) correspond to Wasserstein gradient flows with respect to a conditioned MMD functional. To this end, set $P_{N,q} \coloneqq \{ \frac{1}{N} \sum_{i=1}^{N} \delta_{x_i,q_i} : (x_i, q_i) \in \mathbb{R}^d \times \mathbb{R}^n, (x_i, q_i) \neq (x_j, q_j), i \neq j \}$.

**Theorem 4.** *For given $(p_i, q_i) \in \mathbb{R}^d \times \mathbb{R}^n$, $i = 1, \dots, N$, set $\nu_{N,q} \coloneqq \frac{1}{N} \sum_{i=1}^{N} \delta_{p_i,q_i}$. Let $u = (u_1, ..., u_N) \colon [0, \infty) \to (\mathbb{R}^d)^N$ be a solution of (10), and assume $(u_i(t), q_i) \neq (u_j(t), q_j)$ for $i \neq j$ and all $t > 0$. Then the curve $\gamma_{N,q} \colon (0, \infty) \to \mathcal{P}_2(\mathbb{R}^d)$ defined by*

$$\gamma_{N,q}(t) = \frac{1}{N} \sum_{i=1}^{N} \delta_{u_i(t),q_i}$$

*is a Wasserstein gradient flow with respect to the functional $\mathcal{J}_{\nu_{N,q}} \colon \mathcal{P}_2(\mathbb{R}^d) \to \mathbb{R} \cup \{\infty\}$ given by*

$$\mathcal{J}_{\nu_{N,q}} \coloneqq \begin{cases} \mathcal{F}_{\nu_{N,q}}, & \text{if } \mu \in P_{N,q}, \\ \infty, & \text{otherwise,} \end{cases}$$

*where $\mathcal{F}_{\nu_{N,q}}$ is the functional defined in (2).*

The definition of Wasserstein gradient flows and the proof are included in Appendix A.3. In particular, we will see that there is no flow in the second component.

In order to approximate the joint distribution $P_{X,Y}$ starting in $P_{Z,Y}$, Du et al. (2023) obtained convincing results by considering (discretized) Wasserstein gradient flows with respect to the sliced Wasserstein distance

$$\mathcal{SW}_2^2(\mu, \nu) = \mathbb{E}_{\xi \in \mathcal{S}^{d-1}}[\mathcal{W}_2^2(P_{\xi\#}\mu, P_{\xi\#}\nu)], \quad P_\xi(x) = \langle \xi, x \rangle.$$

They observed in their experiments that there is nearly no flow in the second component, but acknowledged that they *"are unable to provide a rigorous theoretical justification for the time being."* Our proof of Theorem 4, more precisely Corollary 13 in Appendix A.3 delivers the theoretical justification of their empirical result.

### 3.3 CONDITIONAL GENERATIVE MMD FLOWS

In this subsection we want to learn the mapping $T \colon \mathbb{R}^d \times \mathbb{R}^n \to \mathbb{R}^d$ describing the evolution, how the initial particles $(z_i, q_i)$ move to $u_i(t_{\max})$, where $u = (u_1, ..., u_N) \colon [0, \infty) \to \mathbb{R}^{dN}$ solves the ODE (10). To this

end, we adopt the generative MMD flows from Hertrich et al. (2024) and simulate the ODE (10) using an explicit Euler scheme. More precisely, we compute iteratively $u^{(k)} = (u_1^{(k)}, ..., u_N^{(k)})$ by

$$u^{(k+1)} = u^{(k)} - \tau N \nabla_x F_{(p,q)}((u^{(k)}, q)), \quad u_i^{(0)} = z_i. \tag{11}$$

In order to evaluate the gradient efficiently and to speed up the computations, we use the sliced computation of $\nabla_x F_{(p,q)}((u^{(k)}, q))$ and the momentum form of (11) as proposed in Hertrich et al. (2024). Now, we train neural networks $\Phi_1, ..., \Phi_L \colon \mathbb{R}^d \times \mathbb{R}^n \to \mathbb{R}^d$ taking $(u_i, q_i)$ as an input, such that each network approximates a fixed number $T_l$ of explicit Euler steps from (11). The precise training procedure is outlined in Algorithm 1 in the appendix. Once the networks $\Phi_l$ are trained, the approximating mapping $T$ is given by $T(\cdot, y) = \Phi_L(\cdot, y) \circ \cdots \circ \Phi_1(\cdot, y)$. In particular, we can generate samples from the approximated posterior distribution $P_{X|Y=y}$ by drawing $z \sim P_Z$ and computing $T(z, y)$, even for samples $y$ of $Y$ which are not contained in the training set.

## 4 EXPERIMENTS

We apply our conditional generative MMD flows to generate images for given conditions $y$ in two settings, namely i) class-conditional image generation, and ii) reconstruction from posterior distributions $P_{X|Y=y}$ in inverse problems. The chosen networks $(\Phi_l)_{l=1}^L$ are UNets (Ronneberger et al., 2015), where we adopted the implementation from Huang et al. (2021) based on Ho et al. (2020). Further details are given in Appendix C.

### 4.1 CLASS-CONDITIONAL IMAGE GENERATION

We choose the condition $Y$ to be the one-hot vectors of the class labels in order to generate samples of MNIST (LeCun et al., 1998), FashionMNIST (Xiao et al., 2017) and CIFAR10 (Krizhevsky, 2009) for given class labels. Figure 2 illustrates the generated samples and reports the average class conditional FID values. That is, we compute the FID between the generated samples from a specific class with all test samples from the same class. Note that class conditional FID values are not comparable with unconditional FIDs. We compare the results with $\ell$-SWF of Du et al. (2023). We observe that the conditional MMD flow generates samples of good visual quality and outperforms Du et al. (2023). Further examples are given in Appendix D.

### 4.2 INVERSE PROBLEMS

**Inpainting.** For the inpainting task with the mask operator $f$, let $y$ be the partially observed images. Inpainted images of MNIST, FashionMNIST and CIFAR10 are shown in Figure 3. The observed images are the leftmost ones, while the unknown ground truth images are given in the rightmost column. The various generated samples in the middle column have very good reconstruction quality and are in particular consistent with the observed part. Their pixelwise standard deviation (std) is given in the second last column.

**Superresolution.** For image superresolution, we consider the superresolution operator $f$ and low-resolution images $y$. Reconstructed high-resolution images of CelebA (Liu et al., 2015) are illustrated in Figure 3. For CelebA, we centercrop the images to $140 \times 140$ and then bicubicely downsample them to $64 \times 64$. Again, the observations are the leftmost ones, the unknown ground truth the rightmost ones and the reconstructions in the middle column are of good quality and high variety. Another superresolution example on high-dimensional and real-world images of materials' microstructures is presented in Figure 4. We benchmark our conditional MMD flow against a conditional normalizing flow "SRFlow" (Lugmayr et al., 2020) and WPPFlow (Altekrüger & Hertrich, 2023) in terms of PSNR and SSIM (Wang et al., 2004). A more detailed description is given in Appendix E.

**Computed Tomography.** The forward operator $f$ is the discretized linear Radon transform and the noise process is given by a scaled negative log-Poisson noise, for details see, e.g., Altekrüger et al. (2023a);

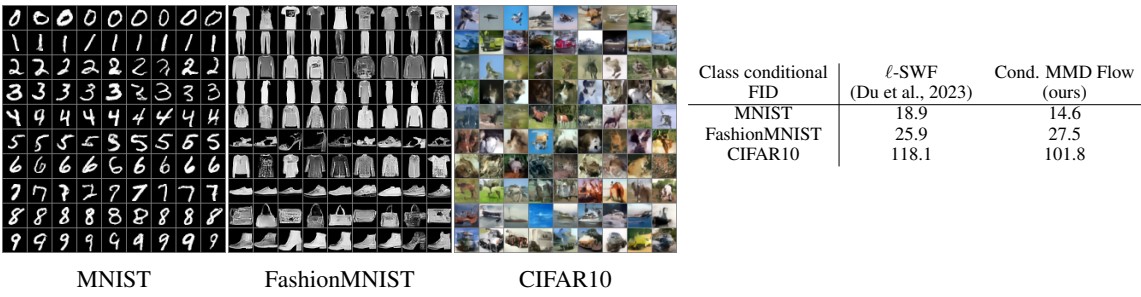

| Class conditional FID | $\ell$-SWF (Du et al., 2023) | Cond. MMD Flow (ours) |
|---|---|---|
| MNIST | 18.9 | 14.6 |
| FashionMNIST | 25.9 | 27.5 |
| CIFAR10 | 118.1 | 101.8 |

MNIST        FashionMNIST        CIFAR10

Figure 2: Class-conditional samples of MNIST, FashionMNIST and CIFAR10 and average class conditional FIDs. Note that these FID values are **not** comparable to unconditional FID values. A more detailed version is given in Table 1.

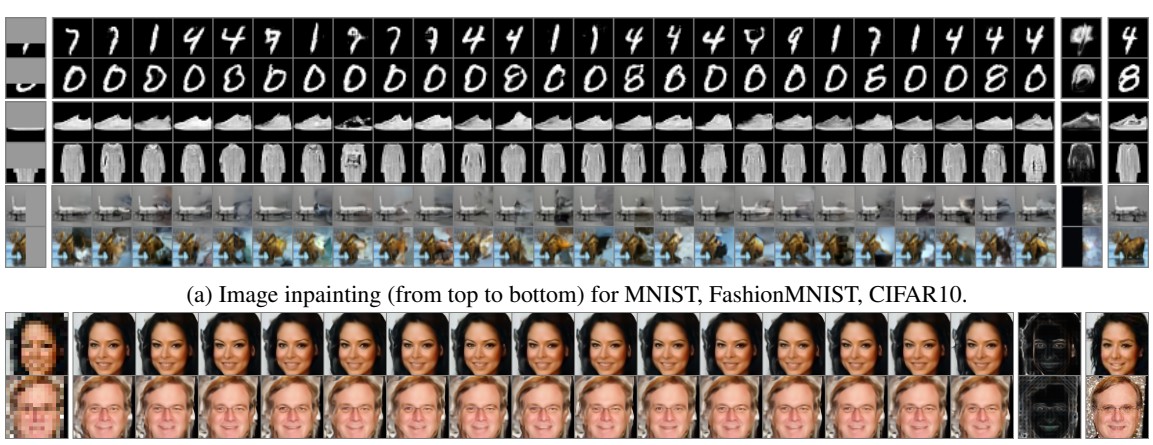

(a) Image inpainting (from top to bottom) for MNIST, FashionMNIST, CIFAR10.

(b) Image superresolution for CelebA with magnification factor 4.

Figure 3: Image inpainting and superresolution for different data sets.

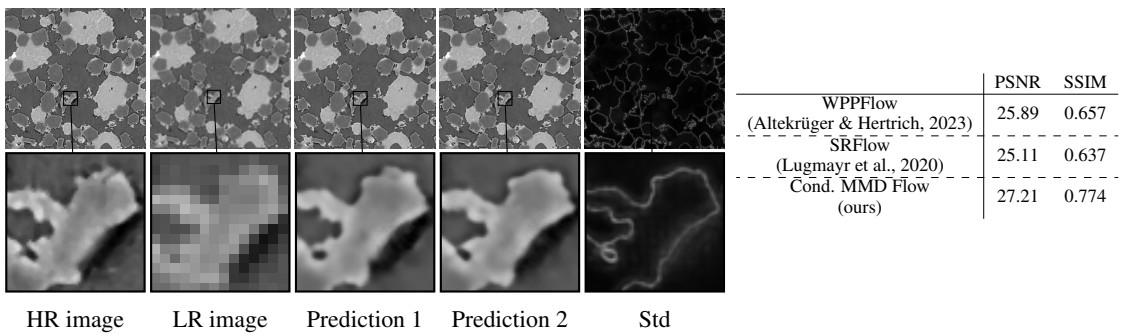

| | PSNR | SSIM |
|---|---|---|
| WPPFlow (Altekrüger & Hertrich, 2023) | 25.89 | 0.657 |
| SRFlow (Lugmayr et al., 2020) | 25.11 | 0.637 |
| Cond. MMD Flow (ours) | 27.21 | 0.774 |

HR image        LR image        Prediction 1        Prediction 2        Std

Figure 4: Two different posterior samples and pixel-wise standard deviation for superresolution using conditional MMD flows.

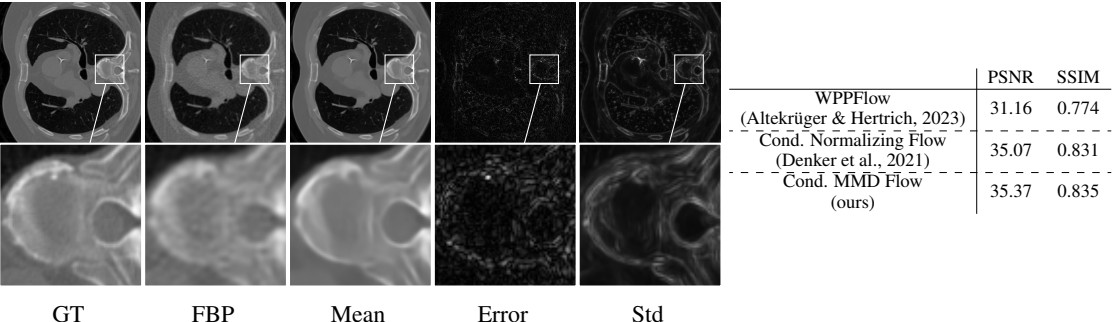

| | | PSNR | SSIM |
|---|---|---|---|
| WPPFlow (Altekrüger & Hertrich, 2023) | | 31.16 | 0.774 |
| Cond. Normalizing Flow (Denker et al., 2021) | | 35.07 | 0.831 |
| Cond. MMD Flow (ours) | | 35.37 | 0.835 |

GT      FBP      Mean      Error      Std

Figure 5: Generated mean image, error towards ground truth and pixel-wise standard deviation for low dose computed tomography using conditional MMD flows.

Leuschner et al. (2021). The data is taken from the LoDoPaB dataset of Leuschner et al. (2021) for *low-dose CT imaging*. The results are shown in Figure 5. We illustrate the mean image of 100 reconstructions, its error towards the ground truth and the standard deviation of the reconstructions. A quantitative comparison with (Denker et al., 2021) for this example is given in Fig 5. Here we provide PSNR and SSIM of the mean images for the whole testset containing 3553 images. More examples towards limited angle CT and low-dose CT are given in Figures 9 and 10 in Appendix D.

## 5 CONCLUSION

We introduced conditional MMD flows with negative distance kernel and applied them for posterior sampling in inverse problems by approximating the joint distribution. To prove stability of our model, we bounded the expected approximation error of the posterior distribution by the error of the joint distribution. We represented our conditional MMD flows as Wasserstein gradient flows, which also provides additional insights for the recent paper by Du et al. (2023). Finally, we applied our algorithm to conditional image generation, inpainting, superresolution and CT. From a theoretical viewpoint it would be interesting to check whether the dimension scaling in Theorem 2 is optimal. In this paper we focused on the negative distance kernel because it can be computed efficiently via slicing and sorting. It would be interesting if similar results hold for other kernels. Moreover, so far we only considered discrete gradient flows with a fixed number $N$ of particles. The convergence properties of these flows in the mean-field limit $N \to \infty$ is only partially answered in the literature, see e.g. (Carrillo et al., 2020) for the one-dimensional setting or (Arbel et al., 2019) for a result with smooth kernels. From a numerical perspective, it could be beneficial to consider more general slicing procedures, see e.g. (Kolouri et al., 2019a; Nguyen & Ho, 2023), or other kernels, see Hertrich (2024).

**Limitations.** It is the aim of our numerical examples to demonstrate that our method can be used for highly ill-posed and high-dimensional imaging inverse problems. In particular, we do *not* claim that our computed tomography experiments are realistic for clinical applications. In practice, the availability and potential biases of high-quality datasets are critical bottlenecks in medical imaging. Moreover, even slight changes in the forward operator or noise model require that the whole model is retrained, which is computational costly and demands a corresponding dataset. Since the particles are interacting, an important next step would be to enable batching to train our model. Finally, we see our work as mainly theoretical but also provide evidence for its scalability to high-dimensional and complicated inverse problems. The precise evaluation of posterior sampling algorithms in high dimensions is very hard due to the lack of meaningful quality metrics.

## ACKNOWLEDGMENTS

P.H. acknowledges funding by the German Research Foundation (DFG) within the project SPP 2298 "Theoretical Foundations of Deep Learning", J.H. within the project STE 571/16-1 and by the EPSRC programme grant "The Mathematics of Deep Learning" with reference EP/V026259/1, F.A. by the DFG under Germany's Excellence Strategy – The Berlin Mathematics Research Center MATH+ (project AA5-6), R.B. and J.C. within the BMBF project 'VI-Screen' (13N15754) and G.S. acknowledges support by the BMBF Project "Sale" (01|S20053B).

The material data from Section 4 has been acquired in the frame of the EU Horizon 2020 Marie Sklodowska-Curie Actions Innovative Training Network MUMMERING (MUltiscale, Multimodal and Multidimensional imaging for EngineeRING, Grant Number 765604) at the beamline TOMCAT of the SLS by A Saadaldin, D Bernard, and F Marone Welford. We acknowledge the Paul Scherrer Institut, Villigen, Switzerland for provision of synchrotron radiation beamtime at the TOMCAT beamline X02DA of the SLS.

We would like to thank Chao Du for providing us class-conditional samples of his method for a quantitative comparison in Figure 2.

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

# A  SUPPLEMENT TO SECTION 3

## A.1  PROOF OF PROPOSITION 1

By the definition of the conditional probability as the disintegration of the joint measure, we have to prove that $P_Y \times k = P_{Y,X}$, with $k(y, \cdot) = T(\cdot, y)_\# P_Z$. To this end, let $A \in \mathcal{B}(\mathbb{R}^n \times \mathbb{R}^d)$ be Borel measurable. Then it holds

$$(P_Y \times k)(A) = \int_{\mathbb{R}^n} \int_{\mathbb{R}^d} 1_A(x,y) \, \mathrm{d}T(\cdot,y)_\# P_Z(x) \, \mathrm{d}P_Y(y)$$

$$= \int_{\mathbb{R}^n} \int_{\mathbb{R}^d} 1_A(T(z,y),y) \, \mathrm{d}P_Z(z) \, \mathrm{d}P_Y(y)$$

Since $Y$ and $Z$ are independent by assumption, this is equal to

$$\int_{\mathbb{R}^n \times \mathbb{R}^d} \int_{\mathbb{R}^d} 1_A(T(z,y),y) \, \mathrm{d}P_{Y,Z}(y,z)$$

$$= \int_{\mathbb{R}^n \times \mathbb{R}^d} \int_{\mathbb{R}^d} 1_A(x,y) \, \mathrm{d}P_{Y,T(Z,Y)}(y,x) = P_{Y,T(Z,Y)}(A) = P_{Y,X}(A),$$

where the first equality follows by the transformation formula and the last equality follows by assumption. Since this holds for all $A \in \mathcal{B}(\mathbb{R}^n \times \mathbb{R}^d)$, this completes the proof. $\qquad\square$

## A.2  SUPPLEMENT TO SUBSECTION 3.1

The proof of Theorem 2 requires some preliminaries on RKHS which can be found more detailed, e.g. in Steinwart & Christmann (2008); Wendland (2004). The negative distance kernel $K(x,y) = -\|x-y\|$ is a conditionally positive definite function meaning that for every $N \in \mathbb{N}$ and $a_i \in \mathbb{R}$ with $\sum_{i=1}^N a_i = 0$ and $x_i \in \mathbb{R}^d$, $i = 1, \ldots, N$, it holds

$$\sum_{i,j=1}^N a_i a_j K(x_i, x_j) \geq 0, \tag{12}$$

where equality is only possible if $a_i = 0$ for all $i = 1, \ldots, N$. The associated kernel

$$\tilde{K}(x,y) := -\|x-y\| + \|x\| + \|y\|$$

is positive definite, i.e., $\tilde{K}$ fulfills (12) without the sum constraint on the $a_i$. For any $\mu, \nu \in \mathcal{P}_1(\mathbb{R}^d)$, we have

$$\mathcal{D}_{\tilde{K}}(\mu, \nu) = \mathcal{D}_K(\mu, \nu).$$

A Hilbert space of functions $\mathcal{H}(\mathbb{R}^d)$ mapping from $\mathbb{R}^d$ to $\mathbb{R}$ with an inner product $\langle \cdot, \cdot \rangle_\mathcal{H}$, associated norm and with the property that point evaluations $f \mapsto f(x)$ are continuous for all $f \in \mathcal{H}(\mathbb{R}^d)$ is called *reproducing kernel Hilbert space* (RKHS). For symmetric, positive definite functions and in particular for $\tilde{K}$, there exists a unique RKHS such that the *reproducing kernel property*

$$f(x) = \langle f, \tilde{K}(x, \cdot) \rangle_\mathcal{H} \quad \text{for all } f \in \mathcal{H}(\mathbb{R}^d)$$

holds true. We denote this RKHS by $\mathcal{H}_{\tilde{K}}(\mathbb{R}^d)$. Further, there is an injective mapping from $\mathcal{P}_{\frac{1}{2}}(\mathbb{R}^d)$ to $\mathcal{H}_{\tilde{K}}(\mathbb{R}^d)$, called *kernel mean embedding* defined by

$$\hat{\mu}(x) := \langle \tilde{K}(x, \cdot), \mu \rangle = \int_{\mathbb{R}^d} \tilde{K}(x,y) \, \mathrm{d}\mu(y),$$

see, e.g., Modeste & Dombry (2023). Note that this has nothing to do with the characteristic function of $\mu$. Since we do not address the later one in this paper, there is no notation mismatch. The kernel mean embedding is not surjective, see Steinwart & Fasciati-Ziegel (2021). We have for $h \in \mathcal{H}_{\tilde{K}}(\mathbb{R}^d)$ and $\mu \in \mathcal{P}_{\frac{1}{2}}(\mathbb{R}^d)$ the representation

$$\langle h, \mu \rangle = \int_{\mathbb{R}^d} h(x) \, \mathrm{d}\mu(x) = \int_{\mathbb{R}^d} \langle h, \tilde{K}(x, \cdot) \rangle_{\mathcal{H}_{\tilde{K}}} \, \mathrm{d}\mu(x)$$

$$= \left\langle h, \int_{\mathbb{R}^d} \tilde{K}(x, \cdot) \, \mathrm{d}\mu(x) \right\rangle_{\mathcal{H}_{\tilde{K}}} = \langle h, \hat{\mu} \rangle_{\mathcal{H}_{\tilde{K}}}. \tag{13}$$

This implies together with the dual representation of the MMD (Novak & Wozniakowski, 2010) for $\mu, \nu \in \mathcal{P}_{\frac{1}{2}}(\mathbb{R}^d)$ that

$$\mathcal{D}_{\tilde{K}}(\mu, \nu) = \sup_{\|h\|_{\mathcal{H}_{\tilde{K}}} \leq 1} \langle h, \mu - \nu \rangle = \sup_{\|h\|_{\mathcal{H}_{\tilde{K}}} \leq 1} \langle h, \hat{\mu} - \hat{\nu} \rangle_{\mathcal{H}_{\tilde{K}}} = \|\hat{\mu} - \hat{\nu}\|_{\mathcal{H}_{\tilde{K}}}. \tag{14}$$

Equipped with the Wasserstein-$p$ distance defined by

$$\mathcal{W}_p(\mu, \nu) := \begin{cases} \left( \inf_{\pi \in \Pi(\mu, \nu)} \int_{\mathbb{R}^d \times \mathbb{R}^d} \|x - y\|^p \, \mathrm{d}\pi(x, y) \right)^{\frac{1}{p}} & \text{if } p \in [1, \infty), \\ \inf_{\pi \in \Pi(\mu, \nu)} \int_{\mathbb{R}^d \times \mathbb{R}^d} \|x - y\|^p \, \mathrm{d}\pi(x, y) & \text{if } p \in (0, 1), \end{cases} \tag{15}$$

where $\Pi(\mu, \nu)$ denotes the set of measures with marginals $\mu$ and $\nu$, the space $\mathcal{P}_p(\mathbb{R}^d)$ becomes a complete and separable metric space, see Modeste & Dombry (2023); Villani (2003). We will only need $p \in \{\frac{1}{2}, 1, 2\}$. Note that by Jensen's inequality,

$$W_{\frac{1}{2}}^2(\mu, \nu) \leq W_1(\mu, \nu) \leq W_2(\mu, \nu),$$

i.e., convergence in $\mathcal{P}_p$ is stronger for larger values of $p$. Finally, we need the dual representation of the Wasserstein-$\frac{1}{2}$ distances

$$W_{\frac{1}{2}}(\mu, \nu) = \sup_{|h|_{\mathcal{C}^{\frac{1}{2}}} \leq 1} \langle h, \mu - \nu \rangle, \tag{16}$$

where $\mathcal{C}^{\frac{1}{2}}(\mathbb{R}^d)$ denotes the space of $\frac{1}{2}$-Hölder continuous functions together with the seminorm $|f|_{\mathcal{C}^{\frac{1}{2}}} := \sup_{x \neq y} (f(x) - f(y))/\|x - y\|$, see Modeste & Dombry (2023); Villani (2003). For compactly supported measures, a relation between the MMD with the negative distance kernel and the Wasserstein distance was proven in Hertrich et al. (2024).

**Lemma 5.** *Let $K(x, y) := -\|x - y\|$. For $\mu, \nu \in \mathcal{P}_1(\mathbb{R}^d)$ it holds*

$$2 \mathcal{D}_K^2(\mu, \nu) \leq W_1(\mu, \nu).$$

*If $\mu$ and $\nu$ are additionally supported on the ball $B_R(0)$, then there exists a constant $C_d > 0$ such that*

$$W_1(\mu, \nu) \leq C_d R^{\frac{2d+1}{2d+2}} \mathcal{D}_K(\mu, \nu)^{\frac{1}{d+1}}.$$

Now we can prove Theorem 2.

**Proof of Theorem 2** 1. For any $y \in \mathbb{R}^n$, we consider the difference of the kernel mean embedding functions $f(\cdot, y) := \hat{P}_{\tilde{X}|Y=y} - \hat{P}_{X|Y=y}$. By (14) and (13), we obtain

$$\mathcal{D}_K^2(P_{\tilde{X}|Y=y}, P_{X|Y=y}) = \langle f(\cdot, y), \hat{P}_{\tilde{X}|Y=y} - \hat{P}_{X|Y=y} \rangle_{\mathcal{H}_{\tilde{K}}} = \langle f(\cdot, y), P_{\tilde{X}|Y=y} - P_{X|Y=y} \rangle$$

and further

$$\mathbb{E}_{y \sim P_Y} \left[ \mathcal{D}_K^2(P_{\tilde{X}|Y=y}, P_{X|Y=y}) \right] = \int_{\mathbb{R}^n} \langle f(\cdot, y), P_{\tilde{X}|Y=y} - P_{X|Y=y} \rangle \, d\mathcal{P}_Y(y)$$

$$= \int_{\mathbb{R}^n} \int_{\mathbb{R}^d} f(x, y)(p_{\tilde{X}|Y=y} - p_{X|Y=y})(x) p_Y(y) \, dx \, dy.$$

Applying the definition of the posterior density $p_{X|Y=y} = \frac{p_{X,Y}}{p_Y}$, this can be rewritten as

$$\mathbb{E}_{y \sim P_Y} \left[ \mathcal{D}_K^2(P_{\tilde{X}|Y=y}, P_{X|Y=y}) \right] = \int_{\mathbb{R}^n} \int_{\mathbb{R}^d} f(x, y)(p_{\tilde{X},Y} - p_{X,Y})(x, y) \, dx \, dy. \tag{17}$$

2. Next, we show that the function $f(x, y)$ is $\frac{1}{2}$-Hölder continuous with respect to both arguments. First, we conclude by assumption on the $\frac{1}{2}$-Hölder continuity of the posteriors with respect to $y$ that

$$f(t, y_1) - f(t, y_2) = \int_{\mathbb{R}^d} \tilde{K}(x, t) \, dP_{\tilde{X}|Y=y_1}(x) - \int_{\mathbb{R}^d} \tilde{K}(x, t) \, dP_{X|Y=y_1}(x)$$

$$- \int_{\mathbb{R}^d} \tilde{K}(x, t) \, dP_{\tilde{X}|Y=y_2}(x) + \int_{\mathbb{R}^d} \tilde{K}(x, t) \, dP_{X|Y=y_2}(x)$$

$$= \int_{\mathbb{R}^d} \tilde{K}(x, t)(p_{\tilde{X}|Y=y_1} - p_{\tilde{X}|Y=y_2})(x) \, dx$$

$$+ \int_{\mathbb{R}^d} \tilde{K}(x, t)(p_{X|Y=y_2} - p_{X|Y=y_1})(x) \, dx$$

$$\leq 2 C_{S_n} \int_{S_d} \tilde{K}(x, t) \, dx \|y_1 - y_2\|^{\frac{1}{2}} \leq 2 C_{S_n} C_{S_d} \|y_1 - y_2\|^{\frac{1}{2}},$$

where $C_{S_d} := \max_{t \in S_d} \int_{S_d} \tilde{K}(x, t) \, dx$. Interchanging the role of $y_1$ and $y_2$ yields

$$|f(t, y_1) - f(t, y_2)| \leq C_S \|y_1 - y_2\|^{\frac{1}{2}}, \quad C_S := 2 C_{S_n} C_{S_d}$$

for all $t \in S_d$ and all $y_1, y_2 \in S_n$. Now the triangle inequality and the relation $|h(x) - h(y)| = |\langle h, \tilde{K}(x, \cdot) - \tilde{K}(y, \cdot) \rangle_{\mathcal{H}_{\tilde{K}}}| \leq \|h\|_{\mathcal{H}_{\tilde{K}}} \|\tilde{K}(x, \cdot) - \tilde{K}(y, \cdot)\|_{\mathcal{H}_{\tilde{K}}}$ for all $h \in \mathcal{H}_{\tilde{K}}$, implies for $x_1, x_2 \in S_d$ and $y_1, y_2 \in S_n$ that

$$|f(x_1, y_1) - f(x_2, y_2)| \leq |f(x_1, y_1) - f(x_1, y_2)| + |f(x_1, y_2) - f(x_2, y_2)|$$

$$\leq C_{S_d} \|y_1 - y_2\|^{\frac{1}{2}} + \|f(\cdot, y_2)\|_{\mathcal{H}_{\tilde{K}}} \|\tilde{K}(x_1, \cdot) - \tilde{K}(x_2, \cdot)\|_{\mathcal{H}_{\tilde{K}}}.$$

By the reproducing kernel property and definition of $\tilde{K}$ we have

$$\|\tilde{K}(x_1, \cdot) - \tilde{K}(x_2, \cdot)\|_{\mathcal{H}_{\tilde{K}}}^2 = \tilde{K}(x_1, x_1) + \tilde{K}(x_2, x_2) - 2\tilde{K}(x_1, x_2) = 2\|x_1 - x_2\|,$$

so that

$$|f(x_1, y_1) - f(x_2, y_2)| \leq C_{S_d} \|y_1 - y_2\|^{\frac{1}{2}} + \sqrt{2} \|f(\cdot, y_2)\|_{\mathcal{H}_{\tilde{K}}} \|x_1 - x_2\|^{\frac{1}{2}}$$

$$\leq \frac{2}{2^{\frac{1}{4}}} \max \left( C_{S_d}, \sqrt{2} \|f(\cdot, y_2)\|_{\mathcal{H}_{\tilde{K}}} \right) \|(x_1, y_1) - (x_2, y_2)\|^{\frac{1}{2}}.$$

Finally, we obtain

$$\|f(\cdot, y_2)\|_{\mathcal{H}_{\tilde{K}}} = \mathcal{D}_K(P_{\tilde{X}|Y=y_2}, P_{X|Y=y_2}) = \sup_{\|h\|_{\mathcal{H}_{\tilde{K}}} \leq 1} \int_{\mathbb{R}^d} h \, d(P_{\tilde{X}|Y=y_2} - P_{X|Y=y_2})$$

$$\leq \sup_{\|h\|_{\mathcal{H}_{\tilde{K}}}\leq 1} \left( |\int_{\mathbb{R}^d} h \; \mathrm{d}P_{\tilde{X}|Y=y_2}| + |\int_{\mathbb{R}^d} h \; \mathrm{d}P_{X|Y=y_2}| \right) \leq 2 \sup_{\|h\|_{\mathcal{H}_{\tilde{K}}}\leq 1} \|h\|_\infty$$

$$\leq 2\tilde{C}_{S_d}$$

where $\|h\|_\infty := \max_{x \in S_d} |h(x)|$ and the last inequality follows by $|h(x)| = |\langle h, \tilde{K}(x,\cdot)\rangle_{\mathcal{H}_{\tilde{K}}}| \leq \|h\|_{\mathcal{H}_{\tilde{K}}} \|\tilde{K}(x,\cdot)\|_{\mathcal{H}_{\tilde{K}}} = \sqrt{2}\|h\|_{\mathcal{H}_{\tilde{K}}} \|x\|^{\frac{1}{2}} \leq \tilde{C}_{S_d}\|h\|_\infty$ for all $x \in S_d$. In summary, we proved for all $x_1, x_2 \in S_d$ ad $y_1, y_2 \in S_n$ that

$$|f(x_1,y_1) - f(x_2,y_2)| \leq \tilde{C} \|(x_1,y_1)-(x_2,y_2)\|^{\frac{1}{2}},$$

where $\tilde{C} := \frac{2}{2^{\frac{1}{4}}} \max\left(C_{S_d}, 2\sqrt{2}\tilde{C}_{S_d}\right)$. Hence, $\frac{1}{\tilde{C}}f$ is $\frac{1}{2}$-Hölder continuous with $|h|_{\mathcal{C}^{\frac{1}{2}}} = 1$.

3. Using Jensen's inequality as well as (16) and (17), we conclude

$$\mathbb{E}_{y\sim P_Y} \left[ \mathcal{D}_K(P_{\tilde{X}|Y=y}, P_{X|Y=y}) \right] \leq \left( \mathbb{E}_{y\sim P_Y} \left[ \mathcal{D}_K^2(P_{\tilde{X}|Y=y}, P_{X|Y=y}) \right] \right)^{\frac{1}{2}}$$
$$\leq \tilde{C}^{\frac{1}{2}} \mathcal{W}_{\frac{1}{2}}(P_{\tilde{X},Y}, P_{X,Y})^{\frac{1}{2}} \leq \tilde{C}^{\frac{1}{2}} \mathcal{W}_1(P_{\tilde{X},Y}, P_{X,Y})^{\frac{1}{4}}.$$

Finally, we apply Lemma 5 to get

$$\mathbb{E}_{y\sim P_Y}[\mathcal{D}_K(P_{\tilde{X}|Y=y}, P_{X|Y=y})] \leq C \, \mathcal{D}_K(P_{\tilde{X},Y}, P_{X,Y})^{\frac{1}{4(d+n+1)}}$$

with appropriate constant $C > 0$. This finishes the proof. $\qquad\square$

The assumption that the random vectors $X$, $\tilde{X}$, and $Y$ map onto compact sets cannot be neglected as the following counterexample shows.

*Example* 6 (Theorem 2 fails for non-compactly supported measures). For fixed $m \in N$ and $0 < \epsilon < 6^{-1}$, we consider the random variables $X$, $\tilde{X}$, and $Y$ on $\mathbb{R}$ with joint distributions $P_{X,Y} := \frac{1}{2}\mathcal{U}_{Q_\epsilon(0,0)} + \frac{1}{2}\mathcal{U}_{Q_\epsilon(m,1)}$ and $P_{\tilde{X},Y} := \frac{1}{2}\mathcal{U}_{Q_\epsilon(m,0)} + \frac{1}{2}\mathcal{U}_{Q_\epsilon(0,1)}$, where $\mathcal{U}_\bullet$ denotes the uniform distribution on the indicated set and $Q_\epsilon(x,y)$ the $\epsilon$-ball around $(x,y)$ with respect to the $\infty$-norm. The assumptions on the conditional densities in Theorem 2 are fulfilled with $C_{S_1} := (1-2\epsilon)^{-1/2}$. Using Lemma 5, we obtain $\mathcal{D}_K(P_{\tilde{X},Y}, P_{X,Y}) \leq 2^{-\frac{1}{2}}$ for all $m \in \mathbb{N}$. Furthermore, we have

$$\mathbb{E}_{y\sim P_Y}[\mathcal{D}_K(P_{\tilde{X}|Y=y}, P_{X|Y=y})] = \frac{1}{2}\mathcal{D}_K(\mathcal{U}_{Q_\epsilon(0)}, \mathcal{U}_{Q_\epsilon(m)}) + \frac{1}{2}\mathcal{D}_K(\mathcal{U}_{Q_\epsilon(m)}, \mathcal{U}_{Q_\epsilon(0)}) \geq \sqrt{m-6\epsilon}.$$

Consequently, there cannot exist a constant $C$ such that (7) holds true for random vectors with arbitrary non compact range.

Estimates similar to (7) can also be established for other divergences.

*Remark* 7 (Relation between joint and conditioned distributions for different distances). i) For the *Kullback–Leibler* (KL) *divergence*, the chain rule (Cover & Thomas, 2006, Thm 2.5.3) implies

$$\mathbb{E}_{y\sim P_Y}[\mathrm{KL}(P_{\tilde{X}|Y=y}, P_{X|Y=y})] = \mathrm{KL}(P_{\tilde{X},Y}, P_{X,Y}).$$

ii) Using Lemma 5 and Jensen's inequality, the estimate (7) can be transferred to the Wasserstein-1 distance. More precisely, under the assumptions in Theorem 2, we have

$$\mathbb{E}_{y\sim P_Y}[W_1(P_{\tilde{X}|Y=y}, P_{X|Y=y})] \leq C' \, \mathbb{E}_{y\sim P_Y}[\mathcal{D}_K(P_{\tilde{X}|Y=y}, P_{X|Y=y})^{\frac{1}{d+1}}]$$
$$\leq C'' \mathcal{D}_K(P_{\tilde{X}|Y=y}, P_{X|Y=y})^{\frac{1}{4(d+n+1)(d+1)}} \leq C \, W_1(P_{\tilde{X},Y}, P_{X,Y})^{\frac{1}{8(d+n+1)(d+1)}}$$

with constants $C, C', C'' > 0$.

Next, we show that the relations (8) and (9) hold true under certain assumptions. We need an auxiliary lemma which was proven for more general kernels by Baptista et al. (2023, Thm 3.2).

**Lemma 8.** *Consider $K(x,y) := -\|x - y\|$, and let $S_d \subset \mathbb{R}^d$ be a compact set, and $Z \in S_d$ be a random variable. For $F, G \in L^2_{P_Z}(S_d, \mathbb{R}^d)$ it holds*

$$\mathcal{D}_K\left(F_\# P_Z, G_\# P_Z\right) \leq \sqrt{2}\, \mathbb{E}_{z \sim P_Z}\left[\|F(z) - G(z)\|\right]^{\frac{1}{2}}.$$

Now we can prove the Hölder estimates.

**Lemma 9** (Stability under Pushforward). *Consider $K(x,y) := -\|x - y\|$, and let $S_n \subset \mathbb{R}^n$ and $S_d \subset \mathbb{R}^d$ be compact sets, and $Z \in S_d$ be a random variable. Further, let $T \colon \mathbb{R}^d \times S_n \to \mathbb{R}^d$ be measurable. If the derivatives are uniform bounded by $\sup_{y \in S_n} \|\nabla_y T(z, y)\| \leq C$ for all $z \in S_d$, then*

$$\mathcal{D}_K\left(T(\cdot, y_1)_\# P_Z, T(\cdot, y_2,)_\# P_Z\right) \leq \sqrt{2C}\, \|y_1 - y_2\|^{\frac{1}{2}} \quad \text{for all} \quad y_1, y_2 \in S_n.$$

*Proof.* For all $y_1, y_2 \in S_n$, Lemma 8 yields

$$\mathcal{D}_K\left(T(\cdot, y_1)_\# P_Z, T(\cdot, y_2)_\# P_Z\right) \leq \sqrt{2}\, \mathbb{E}_{z \sim P_Z}\left[\|T(z, y_1) - T(z, y_2)\|\right]^{\frac{1}{2}}.$$

Further, the second fundamental theorem of calculus implies

$$\|T(z, y_1) - T(z, y_2)\| = \left\|\int_0^1 \nabla_y T(z, y_1 + t(y_2 - y_1))(y_1 - y_2)\, \mathrm{d}t\right\|$$

$$\leq \int_0^1 \|\nabla_y T(z, y_1 + t(y_2 - y_1))\|\, \mathrm{d}t\, \|y_1 - y_2\| \leq C\, \|y_1 - y_2\|.$$

for all $z \in S_d$. Applying this estimate in the above inequality yields the assertion. $\quad\square$

For the stability with respect to the posteriors, there exist sophisticated strategies to obtain optimal bounds, see Garbuno-Inigo et al. (2023). However, for our setting, we can just apply a result from Altekrüger et al. (2023b); Sprungk (2020) on the Wasserstein-1 distance.

**Lemma 10** (Stability of Posteriors). *Consider $K(x,y) := -\|x - y\|$. Let $S_n \subset \mathbb{R}^n$ and $S_d \subset \mathbb{R}^d$ be compact sets, and $X \in S_d$ and $Y \in S_n$ be random variables. Assume that there exists a constant $M$ such that for all $y_1, y_2 \in S_n$ and for all $x \in S_d$ it holds*

$$\left|\log p_{Y|X=x}(y_1) - \log p_{Y|X=x}(y_2)\right| \leq M\, \|y_1 - y_2\|.$$

*Then, there exists a constant $C > 0$, such that for all $y_1, y_2 \in S_n$ we have*

$$\mathcal{D}_K\left(P_{X|Y=y_1}, P_{X|Y=y_2}\right) \leq C\|y_1 - y_2\|^{\frac{1}{2}}.$$

*Proof.* By Lemma 5, we know that $D_K(\mu, \nu) \leq C W_1(\mu, \nu)^{\frac{1}{2}}$. Now we can apply (Altekrüger et al., 2023b, Lem. 3) together with the fact that local Lipschitz continuity on compact sets implies just Lipschitz continuity. $\quad\square$

**Proof of Theorem 3** For $\delta := \mathbb{E}_{y \sim P_Y}[\mathcal{D}_K(T^\varepsilon(\cdot, y)_\# P_Z, P_{X|Y=y})]$, Theorem 2 implies $\delta \leq C\, \varepsilon^{\frac{1}{4(d+n+1)}}$. Adapting the lines of the proof of (Altekrüger et al., 2023b, Thm. 5) with respect to MMD instead of the Wasserstein-1 distance, we obtain

$$\mathcal{D}_K(T^\varepsilon(\cdot, y)_\# P_Z, P_{X|Y=y}) \leq \tilde{D}\, \delta^{\frac{1}{2(n+1)}} \leq D\, \varepsilon^{\frac{1}{8(d+n+1)(n+1)}}$$

for arbitrary $y \in S_n$, where the constants $D, \tilde{D} > 0$ depend on the dimension $n$, the value $p_Y(y)$, the diameter of $S_n$, the constants from (5), (6), (8), and (9) as well as on $C'$ from the assumptions. Finally, taking $\varepsilon \to 0$ yields the assertion. $\quad\square$

A.3 SUPPLEMENT TO SUBSECTION 3.2

A curve $\gamma\colon I \to \mathcal{P}_2(\mathbb{R}^d)$ on the interval $I \subseteq \mathbb{R}$ is called *absolutely continuous* if there exists a Borel velocity field $v_t\colon \mathbb{R}^d \to \mathbb{R}^d$ with $\int_I \|v_t\|_{L_{2,\gamma(t)}}\, dt < \infty$ such that the continuity equation

$$\partial_t \gamma(t) + \nabla \cdot (v_t \gamma(t)) = 0 \tag{18}$$

is fulfilled on $I \times \mathbb{R}^d$ in a weak sense, see (Ambrosio et al., 2005, Thm. 8.3.1). A locally absolutely continuous curve $\gamma\colon (0,\infty) \to \mathcal{P}_2(\mathbb{R}^d)$ with velocity field $v_t \in \mathrm{T}_{\gamma(t)}\mathcal{P}_2(\mathbb{R}^d)$ is a *Wasserstein gradient flow* with respect to a functional $\mathcal{F}\colon \mathcal{P}_2(\mathbb{R}^d) \to (-\infty, \infty]$ if

$$v_t \in -\partial\mathcal{F}(\gamma(t)) \quad \text{for a.e. } t > 0,$$

where $\mathrm{T}_\mu \mathcal{P}_2(\mathbb{R}^d)$ denotes the *regular tangent space* at $\mu \in \mathcal{P}_2(\mathbb{R}^d)$, see (Ambrosio et al., 2005, Def. 8.4.1), and $\partial\mathcal{F}(\mu)$ the *reduced Fréchet subdiffential* at $\mu \in \mathcal{P}_2(\mathbb{R}^d)$ consisting of all $\xi \in L_2(\mathbb{R}^d, \mathbb{R}^d; \mu)$ satisfying

$$\mathcal{F}(\nu) - \mathcal{F}(\mu) \geq \inf_{\pi \in \Gamma^{\mathrm{opt}}(\mu,\nu)} \int_{\mathbb{R}^d \times \mathbb{R}^d} \langle \xi(x), y - x \rangle\, d\pi(x,y) + o(W_2(\mu,\nu)) \quad \text{for all } \nu \in \mathcal{P}_2(\mathbb{R}^d),$$

see (Ambrosio et al., 2005, Eq. (10.3.13)). Here the infimum is taken over the set $\Gamma^{\mathrm{opt}}(\mu,\nu)$ consisting of all optimal Wasserstein-2 transport plans between $\mu$ and $\nu$, i.e., the minimizer of (15) for $p = 2$.

In order to prove Theorem 4, we first prove that an absolutely continuous curve $\gamma$ within the metric space $(\mathcal{P}_2(\mathbb{R}^d \times \mathbb{R}^n), \mathcal{W}_2)$ does not transport mass within the second component, whenever the second marginal is a constant empirical measure. Note that the $i$th marginal can be written as $(P_i)_{\#}\gamma$ using the projections $P_1(x,y) := x$ and $P_2(x,y) := y$.

**Theorem 11.** *Let $\gamma\colon I \to \mathcal{P}_2(\mathbb{R}^d \times \mathbb{R}^n)$ be an absolutely continuous curve with associate vector field $v_t = (v_{t,1}, v_{t,2})\colon \mathbb{R}^d \times \mathbb{R}^n \to \mathbb{R}^d \times \mathbb{R}^n$. If $(P_2)_{\#}\gamma(t)$ is a constant empirical measure $\frac{1}{N}\sum_{i=1}^N \delta_{q_i}$ independent of $t$, then $v_{t,2}$ vanishes $\gamma(t)$-a.e. for almost every $t \in I$.*

*Proof.* Denote by $\pi_t^{t+h} \in \mathcal{P}_2((\mathbb{R}^d \times \mathbb{R}^n) \times (\mathbb{R}^d \times \mathbb{R}^n))$ an arbitrary optimal Wasserstein-2 plan between $\gamma(t)$ and $\gamma(t+h)$, and by $\tilde{P}_i\colon (\mathbb{R}^d \times \mathbb{R}^n)^2 \to (\mathbb{R}^d \times \mathbb{R}^n)$ the projections to the first and second component, i.e. $\tilde{P}_i((x_1, y_1), (x_2, y_2)) := (x_i, y_i)$ for $i \in \{1, 2\}$. For almost every $t \in I$, the associate vector field $v_t$ of $\gamma$ satisfies

$$\lim_{h \to 0} \big(\tilde{P}_1, \tfrac{1}{h}(\tilde{P}_2 - \tilde{P}_1)\big)_{\#}\pi_t^{t+h} = (\mathrm{Id}, v_t)_{\#}\gamma(t), \tag{19}$$

where the left-hand side converges narrowly, see (Ambrosio et al., 2005, Prop 8.4.6). For $n \in \mathbb{N}$, let $f_n\colon [0, \infty) \to [0, 1]$ be a continuous function with $f_n(0) = 0$, $f_n(t) > 0$ for $t \in (0, n)$ and $f_n(t) = 0$ for $t \geq n$. Consider the integral

$$F_n := \int_{(\mathbb{R}^d \times \mathbb{R}^n)^2} f_n(\|y_2\|)\, d\big(\tilde{P}_1, \tfrac{1}{h}(\tilde{P}_2 - \tilde{P}_1)\big)_{\#}\pi_t^{t+h}((x_1, y_1), (x_2, y_2))$$

$$= \int_{(\mathbb{R}^d \times \mathbb{R}^n)^2} f_n(\tfrac{1}{h}\|y_2 - y_1\|)\, d\pi_t^{t+h}((x_1, y_1), (x_2, y_2)). \tag{20}$$

Since $(P_2)_{\#}\gamma(t) = \frac{1}{N}\sum_{i=1}^N \delta_{q_i}$ is a constant empirical measure, every plan $\pi_t^{t+h}$ is supported on

$$\bigcup_{i,j=1}^N (\mathbb{R}^d \times \{q_i\}) \times (\mathbb{R}^d \times \{q_j\}).$$

Hence, on the support of $\pi_t^{t+h}$, the norm $\|y_2 - y_1\|$ in (20) becomes either zero or is bounded by

$$\|y_2 - y_1\| \geq \min\{\|q_i - q_j\| : i \neq j\} =: S.$$

Thus, for $h \leq S/n$, the integral $F_n$ vanishes and the narrow convergence in (19) implies

$$v_{t,2}(x_1, y_1) \notin (-n, 0) \cup (0, n)$$

for $\gamma(t)$-a.e. $(x_1, y_1) \in \mathbb{R}^d \times \mathbb{R}^n$. Since $n \in \mathbb{N}$ is arbitrary, we obtain the assertion. $\qquad\square$

Interestingly, Theorem 11 is in general not true if the second marginal is not an empirical measure as Example 12 below shows. We can now adapt (Altekrüger et al., 2023c, Prop. D.1) for proving Theorem 4.

**Proof of Theorem 4**   Let $\xi := (\xi_1, \ldots, \xi_M) \in \mathbb{R}^{dN}$ satisfy $(\xi_i, q_i) \neq (\xi_j, q_j)$ for all $i \neq j$. Then, there exists an $\epsilon > 0$ such that the optimal transport plan between $\frac{1}{N} \sum_{i=1}^{N} \delta_{(\xi_i, q_i)}$ and $\frac{1}{N} \sum_{i=1}^{N} \delta_{(\eta_i, q_i)}$ is given by $\pi := \frac{1}{N} \sum_{i=1}^{N} \delta_{((\xi_i, q_i),(\eta_i, q_i))}$ for all $\eta \in \mathbb{R}^{dN}$ with $\|\xi - \eta\| \leq \epsilon$. In particular, it follows

$$\mathcal{W}_2^2\left(\frac{1}{N}\sum_{i=1}^{N}\delta_{(\xi_i, q_i)}, \frac{1}{N}\sum_{i=1}^{N}\delta_{(\eta_i, q_i)}\right) = \frac{1}{N}\sum_{i=1}^{N}\|(\xi_i, q_i) - (\eta_i, q_i)\|_2^2 = \frac{1}{N}\sum_{i=1}^{N}\|\xi_i - \eta_i\|_2^2. \quad (21)$$

By an analogous argumentation as in Altekrüger et al. (2023c), $\gamma_{N,q}$ is a locally absolutely continuous curve; and by Theorem 11, the second component of the associate velocity field $v_t = (v_{t,1}, v_{t,2})$ vanishes, i.e. $v_{t,2} \equiv 0$ for almost every $t \in (0, \infty)$. Exploiting (Ambrosio et al., 2005, Prop. 8.4.6), we obtain

$$
\begin{aligned}
0 &= \lim_{h \to 0} \frac{\mathcal{W}_2^2(\gamma_{N,q}(t+h)), (\mathrm{Id} + h v_t)_\# \gamma_{N,q}(t))}{|h|^2} \\
&= \lim_{h \to 0} \frac{\mathcal{W}_2^2\left(\frac{1}{N}\sum_{i=1}^{N}\delta_{(u_i(t+h), q_i)}, \frac{1}{N}\sum_{i=1}^{N}\delta_{(u_i(t) + h v_{t,1}(u_i(t), q_i), q_i)}\right)}{|h|^2} \\
&= \lim_{h \to 0} \frac{1}{N}\sum_{i=1}^{N}\left\|\frac{u_i(t+h) - u_i(t)}{h} - v_{t,1}(u_i(t), q_i)\right\|^2 = \frac{1}{N}\sum_{i=1}^{N}\|\dot{u}_i(t) - v_{t,1}(u_i(t), q_i)\|^2
\end{aligned}
$$

for a.e. $t \in (0, \infty)$, where the first equality in the last line follows from (21). In particular, this implies $\dot{u}_i(t) = v_{t,1}(u_i(t), q_i)$ a.e. such that $N \nabla_x F_{(p,q)}(u(t), q) = (v_{t,1}(u_1(t), q_1), \ldots, v_{t,1}(u_N(t), q_N))$.

For fixed $t$, we now consider an $\epsilon$-ball around $\gamma_{N,q}(t)$ where the Wasserstein-2 optimal transport between $\gamma_{N,q}(t)$ and a measure from $\mu \in P_{N,q}$ becomes unique as discussed in the beginning of the proof. More precisely, the unique plan between $\gamma_{N,q}(t)$ and $\mu := \frac{1}{N}\sum_{i=1}^{N}\delta_{(\eta_i, q_i)}$ with $\mathcal{W}_2(\mu, \gamma_{N,q}(t)) \leq \epsilon$ is then given by $\pi = \frac{1}{N}\sum_{i=1}^{N}\delta_{((u_i(t), q_i),(\eta_i, q_i))}$. Since $u$ is a solution of (10), we obtain

$$
\begin{aligned}
0 &\leq F_{(p,q)}((\eta, q)) - F_{(p,q)}((u(t), q)) + \langle \nabla_x F_{(p,q)}((u(t), q)), \eta - u(t)\rangle + o(\|\eta - u(t)\|) \\
&= \mathcal{J}_{\nu_{N,q}}(\mu) - \mathcal{J}_{\nu_{N,q}}(\gamma_{N,q}(t)) + \frac{1}{N}\sum_{i=1}^{N}\langle v_t(u_i(t), q_i), (\eta_i, q_i) - (u_i(t), q_i)\rangle + o(\mathcal{W}_2(\mu, \gamma_{N,q}(t))) \\
&= \mathcal{J}_{\nu_{N,q}}(\mu) - \mathcal{J}_{\nu_{N,q}}(\gamma_{N,q}(t)) + \int_{(\mathbb{R}^d \times \mathbb{R}^n)^2} \langle v_t(x_1, y_1), (x_2, y_2) - (x_1, y_1)\rangle \, \mathrm{d}\pi((x_1, y_1),(x_2, y_2)) \\
&\quad + o(\mathcal{W}_2(\mu, \gamma_{N,q}(t))).
\end{aligned}
$$

Since $\pi$ is the unique plan in $\Gamma^{\mathrm{opt}}(\gamma_{N,q}(t), \mu)$, and since $\mathcal{J}_{\nu_{N,q}}(\mu) = \infty$ for $\mu \notin P_{N,q}$, we have $v_t \in -\partial \mathcal{J}_{\nu_{N,q}}(\gamma(t))$ showing the assertion. $\qquad\square$

Finally, we construct an explicit example showing that the restriction to empirical second marginals in Theorem 11 is inevitable.

*Example* 12 (Theorem 11 fails for arbitrary marginals). Let $f\colon I \to [0,1]$ be a continuously differentiable function on the interval $I \subseteq \mathbb{R}$, and consider the curve $\gamma_f\colon I \to \mathcal{P}_2(\mathbb{R})$ given by

$$\gamma_f(t) := (1 - f(t))\,\delta_0 + f(t)\,\lambda_{[-1,0]}.$$

Figuratively, $f$ controls how the mass on the interval $[-1,0]$ flows into or out of the point measure located at zero. In order to show that such curves are absolutely continuous, we exploit the associate quantile functions. More generally, for $\mu \in \mathcal{P}_2(\mathbb{R})$, the quantile function is defined as

$$Q_\mu(s) := \min\{x \in \mathbb{R} : \mu((-\infty, x]) \geq s\}, \quad s \in (0, 1).$$

For our specific curve, the quantile functions are piecewise linear and given by

$$Q_{\gamma_f(t)}(s) = \min\{(f(t))^{-1}\,s - 1, 0\},$$

see for instance Hertrich et al. (2023a, Prop 1). Due to the relation between the quantile function and the Wasserstein distance (Villani, 2003, Thm 2.18), we obtain

$$\mathcal{W}_2^2(\gamma_f(t_1), \gamma_f(t_2)) = \int_0^1 |Q_{\gamma_f(t_1)}(s) - Q_{\gamma_f(t_2)}(s)|^2\,\mathrm{d}s$$

$$\leq \int_0^1 |(f(t_1))^{-1}\,s - (f(t_2))^{-1}\,s|^2\,\mathrm{d}s = \tfrac{1}{3}\,|(f(t_1))^{-1} - (f(t_2))^{-1}|^2.$$

If the derivative of $1/f$ is bounded by $M$ on $I$, the mean value theorem yields

$$\mathcal{W}_2(\gamma_f(t_1), \gamma_f(t_2)) \leq \tfrac{M}{\sqrt{3}}\,|t_1 - t_2|,$$

such that $\gamma_f$ is indeed absolutely continuous. Based on the curves $\gamma_f$ on the line, we now consider the curve $\gamma\colon [1/4, 3/4] \to \mathcal{P}_2(\mathbb{R} \times \mathbb{R})$ given by

$$\gamma(t) := \frac{1}{2}\Big((1 - t)\,\delta_{(0,0)} + t\,\lambda_{\{0\}\times[-1,0]} + t\,\delta_{(2,0)} + (1 - t)\,\lambda_{\{2\}\times[-1,0]}\Big),$$

where $\lambda_A$ denotes the uniform measure on the set $A \subseteq \mathbb{R}^2$. Restricting the transport between $\gamma(t_1)$ and $\gamma(t_2)$ along the lines segments $\{0\} \times [-1,0]$ and $\{2\} \times [-1,0]$, and using the above considerations, where the derivatives are bounded by $M = 4$, we obtain

$$\mathcal{W}_2(\gamma(t_1), \gamma(t_2)) \leq \tfrac{4}{\sqrt{3}}\,|t_1 - t_2|, \quad t_1, t_2 \in [1/4, 3/4].$$

Thus $\gamma$ is absolutely continuous too. Furthermore, both marginals $(\pi_1)_{\#}\gamma(t) = \frac{1}{2}(\delta_0 + \delta_2)$ and $(\pi_2)_{\#}\gamma(t) = \frac{1}{2}(\delta_0 + \lambda_{[-1,0]})$ are independent of $t$. If Theorem 11 would hold true for non-empirical marginals, the associate vector field $v_t\colon \mathbb{R}^2 \to \mathbb{R}^2$ has to be the zero everywhere implying that $\gamma$ is constant. Since this is not the case, we would obtain a contradiction.

**Theoretical justification of a numerical observation in Du et al. (2023)** By Ambrosio et al. (2005, Thm. 8.3.1), absolutely continuous curves $\gamma\colon I \to \mathcal{P}_2(\mathbb{R}^d)$ in $\mathcal{P}_2(\mathbb{R}^d)$ correspond to weak solutions of the continuity equation (18). For the sliced Wasserstein gradient flows with target measure $\nu$, an analytic representation of $v_t$ was derived in Bonnotte (2013) as

$$v_t(x) = \mathbb{E}_{\xi \in \mathbb{S}^{d-1}}[\psi'_{t,\xi}(P_\xi(x))\xi], \quad P_\xi(x) = \langle \xi, x \rangle,$$

where $\psi_{t,\xi}$ is the Kantorovic potential between $P_{\xi\#}\gamma(t)$ and $P_{\xi\#}\nu$. In the case that $\gamma(0) = \frac{1}{N}\sum_{i=1}^N \delta_{z_i}$ and $\nu = \frac{1}{M}\sum_{i=1}^M \delta_{p_i}$, this implies that $\gamma(t) = \frac{1}{N}\sum_{i=1}^N \delta_{u_i(t)}$, where $u = (u_1, ..., u_N)$ is a solution of

$$\dot{u}(t) = v_t(u(t)) = \nabla G_p(u(t)), \quad G_p(x) = \mathcal{SW}_2^2\Big(\frac{1}{N}\sum_{i=1}^N \delta_{x_i}, \frac{1}{M}\sum_{j=1}^M \delta_{p_j}\Big).$$

In the context of posterior sampling, Du et al. (2023) considered sliced Wasserstein gradient flows starting at $\gamma(0) = \frac{1}{N} \sum_{i=1}^N \delta_{(z_i, \tilde{q}_i)}$ with target measure $\nu_{M,q} = \frac{1}{M} \sum_{j=1}^M \delta_{(p_i, q_i)}$ such that $\nu_{M,q} \approx P_{X,Y}$ and $\gamma(0) \approx P_Z \times P_Y$ for continuous random variables $X$ and $Y$ and a latent variable $Z$. Then, they observed numerically that the second part $v_{t,2}$ of $v_t = (v_{t,1}, v_{t,2}) \colon \mathbb{R}^d \times \mathbb{R}^n \to \mathbb{R}^d \times \mathbb{R}^n$ is "almost zero". In order to apply Proposition 1, they set the component $v_{t,2}$ in their simulations artificially to zero, which corresponds to solving the ODE

$$\dot{u}(t) = \nabla_x G_{(p,q)}(u(t), \tilde{q}). \tag{22}$$

Du et al. (2023) write by themselves that they are unable to provide a rigorous theoretical justification of the functionality of their algorithm. Using an analogous proof as for Theorem 4, we can now provide this justification as summarized in the following corollary. In particular, the simulations from Du et al. (2023) are still Wasserstein gradient flows, but with respect to a different functional which has the minimizer $\nu_{M,q} \approx P_{X,Y}$.

**Corollary 13.** *Let $u = (u_1, ..., u_N) \colon [0, \infty) \to (\mathbb{R}^d)^N$ be a solution of (22) and assume that $(u_i(t), \tilde{q}_i) \neq (u_j(t), \tilde{q}_j)$ for $i \neq j$ and all $t > 0$. Then, the curve $\gamma_{N,\tilde{q}} \colon (0, \infty) \to \mathcal{P}_2(\mathbb{R}^d)$ defined by*

$$\gamma_{N,\tilde{q}}(t) = \frac{1}{N} \sum_{i=1}^N \delta_{u_i(t), \tilde{q}_i}$$

*is a Wasserstein gradient flow with respect to the functional*

$$\mu \mapsto \begin{cases} \mathcal{SW}_2^2(\mu, \nu_{M,q}), & \text{if } \mu \in P_{N,\tilde{q}} \\ \infty, & \text{otherwise.} \end{cases}$$

## B  ALGORITHM SUMMARY

Here we summarize the training of our conditional MMD Flows, see Algorithm 1.

## C  IMPLEMENTATION DETAILS

The code is written in PyTorch (Paszke et al., 2019) and is available online[1].

We use UNets $(\Phi)_{l=1}^L$[2] which are trained using Adam (Kingma & Ba, 2015) with a learning rate of 0.0005. Since the differences between the particles $x^{(k+1)}$ and $x^{(k)}$ are large when starting simulating (11) and smaller for larger step $k$, we increase the number of simulation steps $T_l$ up to a predefined maximal number of iteration steps $T_{max} = 30000$.

For our experiments, we make use of several improvements, which are explained in the following.

**Sliced MMD equals MMD**   Instead of computing the derivative of the MMD functional in (11) directly, we use the sliced version of MMD shown in Hertrich et al. (2024). More precisely, let $x := (x_1, \dots, x_N) \in (\mathbb{R}^d)^N$, $p := (p_1, \dots, p_M) \in (\mathbb{R}^d)^M$ and let $F_p^d(x) := F_p(x)$ be the discrete MMD functional, where we explicitly note the dependence on the dimension $d$. Then we can rewrite the gradient of the MMD $\nabla_{x_i} F_p^d(x)$ with the negative distance kernel as

$$\nabla_{x_i} F_p^d(x) = c_d \mathbb{E}_{\xi \sim \mathcal{U}_{\mathbb{S}^{d-1}}} [\partial_i F_{\tilde{p}_\xi}^1(\langle \xi, x_1 \rangle, ..., \langle \xi, x_N \rangle) \xi],$$

---

[1] https://github.com/FabianAltekrueger/Conditional_MMD_Flows
[2] modified from https://github.com/hojonathanho/diffusion/blob/master/diffusion_tf/models/unet.py

---

**Algorithm 1** Training of conditional MMD flows

---

**Input:** Joint samples $p_1, ..., p_N, q_1, ..., q_N$ from $P_{X,Y}$, initial samples $u_1^0, ..., u_N^0$, momentum parameters $m_l \in [0, 1)$ for $l = 1, ..., L$.
Initialize $(v_1, ..., v_N) = 0$.
**for** $l = 1, ..., L$ **do**
    - Set $(\tilde{u}_1^{(0)}, ..., \tilde{u}_N^{(0)}) = (u_1^{(l-1)}, ..., u_N^{(l-1)})$.
    - Simulate $T_l$ steps of the (momentum) MMD flow:
    **for** $t = 1, ..., T_l$ **do**
        - Update $v$ by

$$(v_1, ..., v_N) \leftarrow \nabla_x F_{d+n}((\tilde{u}_i^{(k)}, q_i)_{i=1}^N | (p_i, q_i)_{i=1}^N) + m_l(v_1, ..., v_N)$$

        - Update the flow samples:

$$\tilde{u}^{(k+1)} = \tilde{u}^{(k)} - \tau N(v_1, ..., v_N)$$

    **end for**
    - Train $\Phi_l$ such that $\tilde{u}^{(T_l)} \approx \tilde{u}_i^{(0)} - \Phi_l(\tilde{u}_i^{(0)}, q_i)$ by minimizing the loss

$$\mathcal{L}(\theta_l) = \frac{1}{N} \sum_{i=1}^N \|\Phi_l(\tilde{u}_i^{(0)}, q_i) - (\tilde{u}_i^{(0)} - \tilde{u}_i^{(T_l)})\|^2.$$

    - Set $(u_1^{(l)}, ..., u_N^{(l)}) = (u_1^{(l-1)}, ..., u_N^{(l-1)}) - (\Phi_l(u_1^{(l-1)}, q_1), ..., \Phi_l(u_N^{(l-1)}, q_N))$.
**end for**

---

where $\tilde{p}_\xi := (\langle \xi p_1 \rangle, \ldots, \langle \xi, p_M \rangle)$ and $c_d$ is a constant given by

$$c_d := \frac{\sqrt{\pi}\Gamma(\frac{d+1}{2})}{\Gamma(\frac{d}{2})}.$$

Thus it suffices to compute the gradient w.r.t $F_{\tilde{p}_\xi}^1$, which can be done in a very efficient manner, see Hertrich et al. (2024, Section 3).

**Pyramidal schedules** In order to obtain fast convergence of the particle flow (11) even in high dimensions, we make use of a *pyramidal schedule*. The key idea is to simulate the particle flow on different resolutions of the image, from low to high sequentially. Given the target images $p_i \in \mathbb{R}^d$, where $d = C \cdot H \cdot W$ with height $H$, width $W$ and $C$ channels for $i = 1, ..., N$, we downsample the image by a factor $S$. Then we start simulating the ODE (11) with initial particles $x^{(0)} \in \mathbb{R}^{\frac{d}{S^2}}$, i.e., we simulate the ODE in a substantially smaller dimension. After a predefined number of steps $t$, we upsample the current iterate $x^{(t)}$ to the higher resolution and add noise onto it in order to increase the intrinsic dimension of the images. Then we repeat the procedure until the highest resolution is attained.

**Locally-connected projections** Motivated by Du et al. (2023); Nguyen & Ho (2022), the uniformly sampled projections $\xi \in \mathbb{S}^{d-1}$ can be interpreted as a fully-connected layer applied to the vectorized image. Instead, for image tasks the use of locally-connected projections greatly improves the efficiency of the corresponding particle flow. More concretely, for a given patch size $s$ we sample local projections $\xi_\ell$ uniformly from $\mathbb{S}^{cs^2-1}$. Then, we randomly choose a pair $(h, w)$ and extract a patch $E_{(h,w)}(p)$ of our given image $p \in \mathbb{R}^d$, where $(h, w)$ is the upper left corner of the patch and $E_{(h,w)} \colon \mathbb{R}^d \to \mathbb{R}^{cs^2}$ is the correspond-

ing patch extractor. Using this procedure, we simulate the ODE (11) for $E_{(h,w)}(x_i)$ and target $E_{(h,w)}(p_i)$, $i = 1, ..., N$, where the location of the patch is randomly chosen in each iteration.

In order to apply the locally-connected projections on different resolutions, we upsample the projections to different scales, similar to Du et al. (2023) and, depending on the condition $q_i$, locally-connected projections are also used here. Note that here we introduced an inductive bias, since we do not sample uniformly from $\mathbb{S}^{d-1}$ anymore, but it empirically improves the performance of the proposed scheme. A more detailed discussion can be found in (Du et al., 2023; Nguyen & Ho, 2022).

### C.1 CLASS-CONDITIONAL IMAGE GENERATION

For MNIST and FashionMNIST we use $N = 20000$ target pairs $(p_i, q_i)$, $i = 1, ..., N$, where the conditions $q_i$ are the one-hot vectors of the class labels. We use $P = 500$ projections for each scale of the locally-connected projections, for the observation part we use fully-connected projections. For MNIST, we use the patch size $s = 5$ and apply the projections on resolutions $5, 10, 15, 20$ and $25$. For FashionMNIST, the patch size $s = 9$ is used on resolutions $9$ and $27$. In both cases, the networks are trained for $5000$ iterations with a batch size of $100$.

For CIFAR10, we use $N = 40000$ target pairs and make use of the pyramidal schedule, where we first downsample by a factor 8 to resolution $4 \times 4$ and then upsample the iterates $x^{(t)}$ by a factor of 2 after every $700000$ iterations. In the first two resolutions we use $P = 500$ projections and in the third resolution we use $P = 778$ projections. On the highest resolution of $32 \times 32$, we make use of locally-connected projections, where we choose the patch size $s = 7$ on resolutions $7, 14, 21$ and $28$. Here we use $P = 400$ projections and train the networks for $4000$ iterations with a batch size of $100$.

### C.2 INPAINTING

For MNIST and FashionMNIST we use $N = 20000$ target pairs $(p_i, q_i)$, $i = 1, ..., N$, where the conditions $q_i$ are the observed parts of the image. We use again $P = 500$ projections for each scale of the locally-connected projections, for the observation part we use fully-connected projections. For MNIST, we use the patch size $s = 5$ and apply the projections on resolutions $5, 10, 15$ and $20$. For FashionMNIST, the patch size $s = 7$ is used on resolutions $7, 14$ and $21$. In both cases, the networks are trained for $4000$ iterations with a batch size of $100$. For CIFAR10, we use $N = 30000$ target pairs and make use of the same pyramidal schedule as in the class-conditional part, but we increase the resolution after every $600000$ iterations.

### C.3 SUPERRESOLUTION

We use $N = 20000$ target pairs of CelebA, where the low-resolution images are bicubicely downsampled to resolution $16 \times 16$. Similarly to the pyramidal approach for CIFAR10, we downsample the particles by a factor of 8 and increase the resolution after every $600000$ iterations by a factor of 2. While for the first 2 resolutions we use fully-connected projections with $P = 500$ and $P = 768$ projections, for resolutions $32 \times 32$ and $64 \times 64$ we make use of locally-connected projections with $P = 500$ and $s = 7$ on resolutions 7 and 21 for $32 \times 32$ and $7, 21$ and $49$ for $64 \times 64$. For the observations we use fully-connected projections for all resolutions. The networks are trained for $5000$ iterations and a batch size of $10$.

### C.4 COMPUTED TOMOGRAPHY

We use the first $N = 400$ target pairs of the LoDoPaB training set of size $362 \times 362$, where the observations are the FBP reconstructions of the observed sinograms. We use $P = 500$ projections for each scale of the locally-connected projections with a patch size $s = 15$ and use all resolutions between 15 and 135 as well as resolution 270.

Table 1: FID scores of the class-conditional samples for MNIST, FashionMNIST and CIFAR10. Separated for each class.

| MNIST | | | FashionMNIST | | | CIFAR10 | | |
|---|---|---|---|---|---|---|---|---|
| FID | $\ell$-SWF (Du et al., 2023) | Cond. MMD Flow (ours) | FID | $\ell$-SWF (Du et al., 2023) | Cond. MMD Flow (ours) | FID | $\ell$-SWF (Du et al., 2023) | Cond. MMD Flow (ours) |
| Class 0 | 16.3 | 13.2 | Class 0 | 27.9 | 30.4 | Class 0 | 127.6 | 98.8 |
| Class 1 | 16.0 | 23.3 | Class 1 | 16.0 | 22.9 | Class 1 | 126.9 | 100.9 |
| Class 2 | 19.0 | 17.2 | Class 2 | 26.2 | 28.0 | Class 2 | 125.1 | 117.9 |
| Class 3 | 13.8 | 12.8 | Class 3 | 28.3 | 27.3 | Class 3 | 141.0 | 99.4 |
| Class 4 | 19.7 | 13.7 | Class 4 | 25.5 | 23.6 | Class 4 | 103.9 | 99.7 |
| Class 5 | 34.0 | 13.7 | Class 5 | 27.7 | 31.1 | Class 5 | 126.2 | 112.2 |
| Class 6 | 18.9 | 13.2 | Class 6 | 28.5 | 29.1 | Class 6 | 109.6 | 87.1 |
| Class 7 | 14.6 | 14.3 | Class 7 | 21.3 | 23.6 | Class 7 | 112.5 | 109.3 |
| Class 8 | 19.4 | 13.0 | Class 8 | 37.2 | 38.6 | Class 8 | 100.6 | 94.0 |
| Class 9 | 17.5 | 12.0 | Class 9 | 20.7 | 20.0 | Class 9 | 107.9 | 99.3 |
| Average | 18.9 | 14.6 | Average | 25.9 | 27.5 | Average | 118.1 | 101.8 |

| | WPPFlow (mean img) | SRFlow (mean img) | Sliced MMD Flow (mean img) |
|---|---|---|---|
| PSNR | 25.89 (26.92) | 25.11 (27.36) | **27.21** (27.93) |
| SSIM (Wang et al., 2004) | 0.657 (0.766) | 0.637 (0.771) | **0.774** (0.790) |

Table 2: Comparison of superresolution results of different reconstruction schemes. The values are meaned of 100 reconstructions for each observation. The value of the resulting mean image is in brackets. The best value is marked in bold.

## D    FURTHER NUMERICAL EXAMPLES

Here we provide additional generated samples for the experiments from Section 4. Moreover, the FID scores for the class-conditonal image generation is given in Table 1 for each class separately. Note that the arising values are not comparable with unconditional FID values. Obviously, we outperform the $\ell$-SWF of Du et al. (2023), which is the conceptually closest method.

## E    SUPERRESOLUTION ON MATERIAL DATA

In addition to the superresolution example in Section 4, we provide an example for superresolution on real-world material data. The forward operator $f$ consists of a blur operator with a $16 \times 16$ Gaussian blur kernel with standard deviation 2 and a subsampling with stride 4. The low-resolution images used for reconstruction are generated by artificially downsampling and adding Gaussian noise with standard deviation 0.01. We train the conditional MMD flow with 1000 pairs of high- and low-resolution images of size $100 \times 100$ and $25 \times 25$, respectively.

In Figure 11 we consider an unknown ground truth of size $600 \times 600$ and an observation of size $150 \times 150$. We compare the conditional MMD flow with WPPFlow (Altekrüger & Hertrich, 2023) and SRFlow (Lugmayr et al., 2020). Note that while the conditional MMD flow and the SRFlow are trained under the same setting, the WPPFlow just requires the 1000 observations and the exact knowledge of the forward operator and the noise model for training. We can observe that the conditional MMD flow is able to generate sharper and much more realistic reconstructions for the given low-resolution observation. More examples are given in Figure 12. A quantitative comparison of the methods is given in Table 2. Here we consider 100 unseen test observations and reconstruct 100 samples from the posterior for each observation.

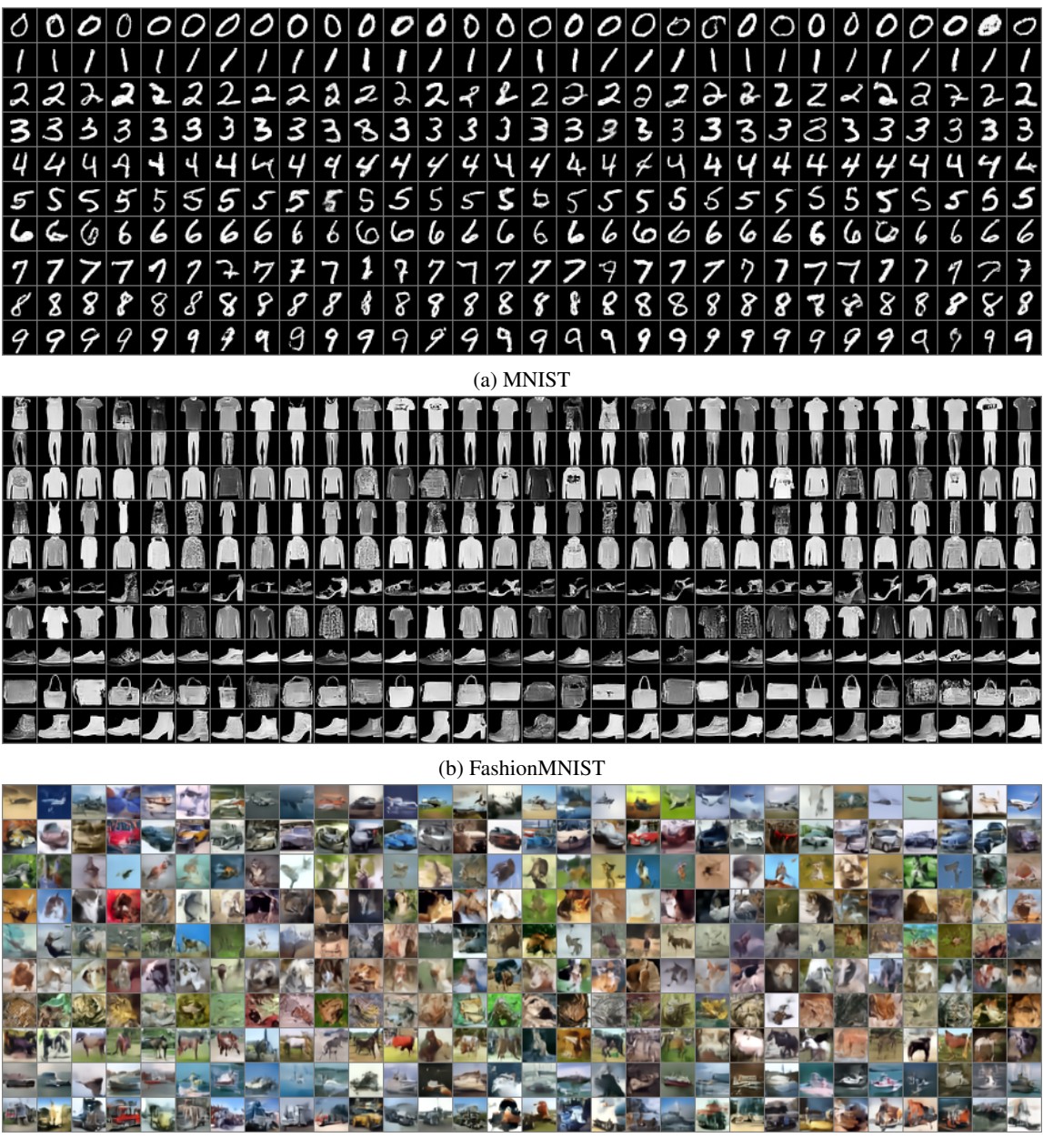

(a) MNIST

(b) FashionMNIST

(c) CIFAR10

Figure 6: Additional class-conditional samples of MNIST, FashionMNIST and CIFAR10.

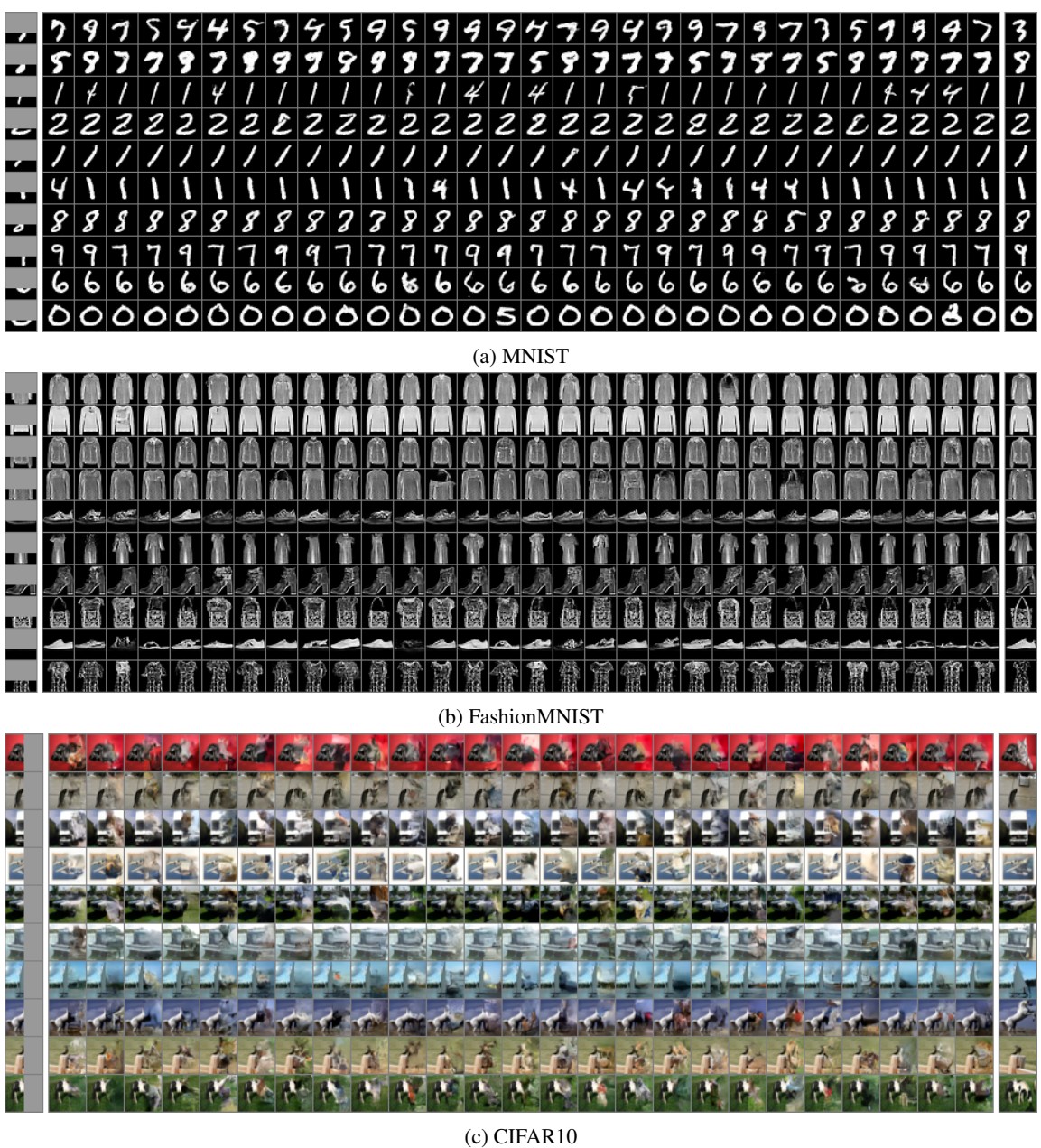

(a) MNIST

(b) FashionMNIST

(c) CIFAR10

Figure 7: Additional inpainted samples of MNIST, FashionMNIST and CIFAR10.

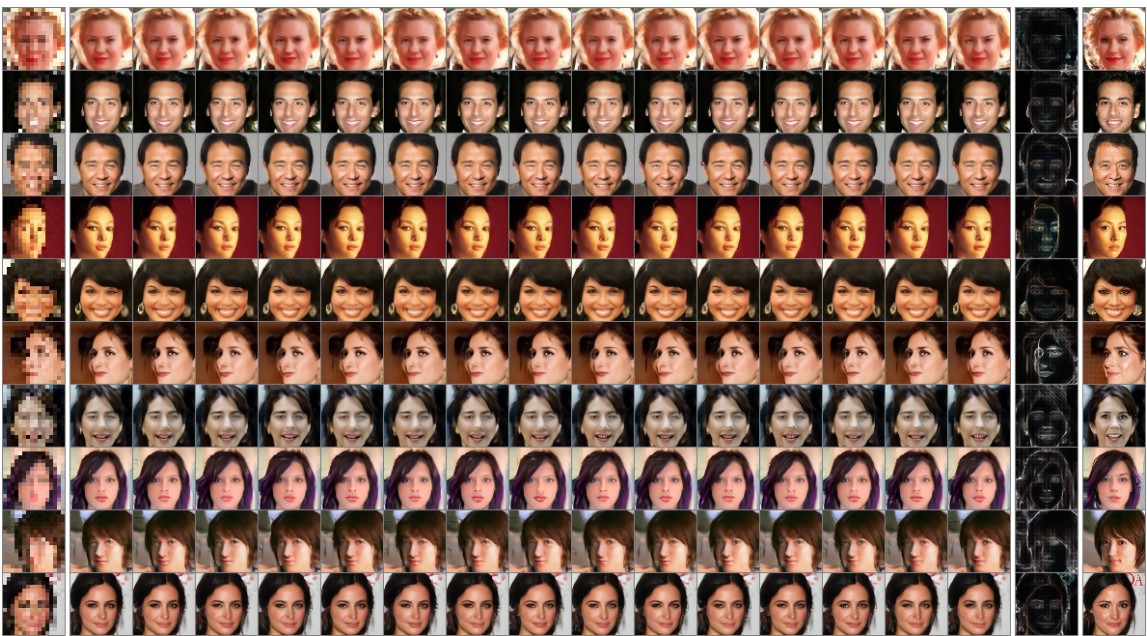

(a) CelebA superresolution

Figure 8: Additional superresoluted images of CelebA

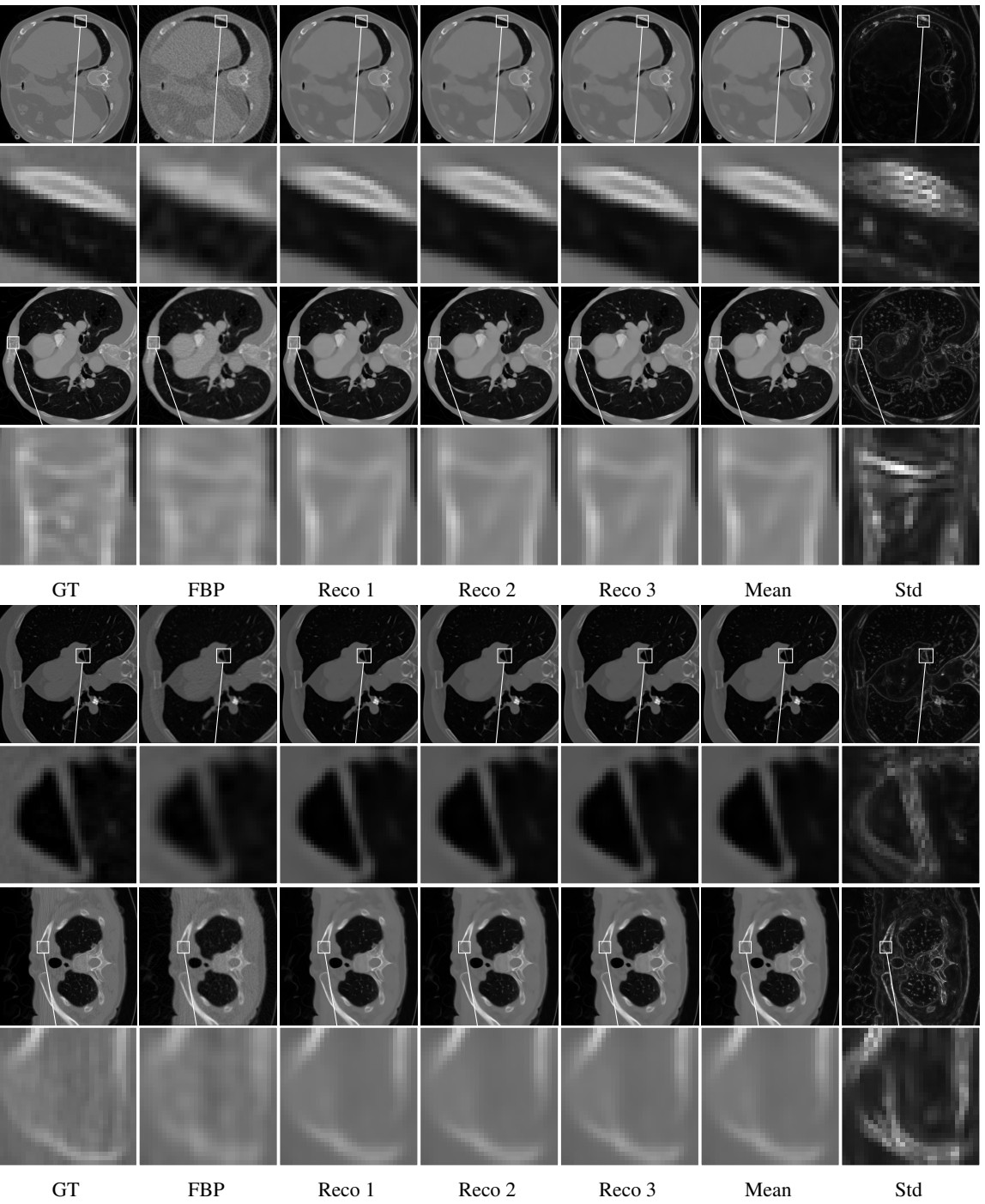

Figure 9: Additional generated posterior samples, mean image and pixel-wise standard deviation for low-dose computed tomography using conditional MMD flows.

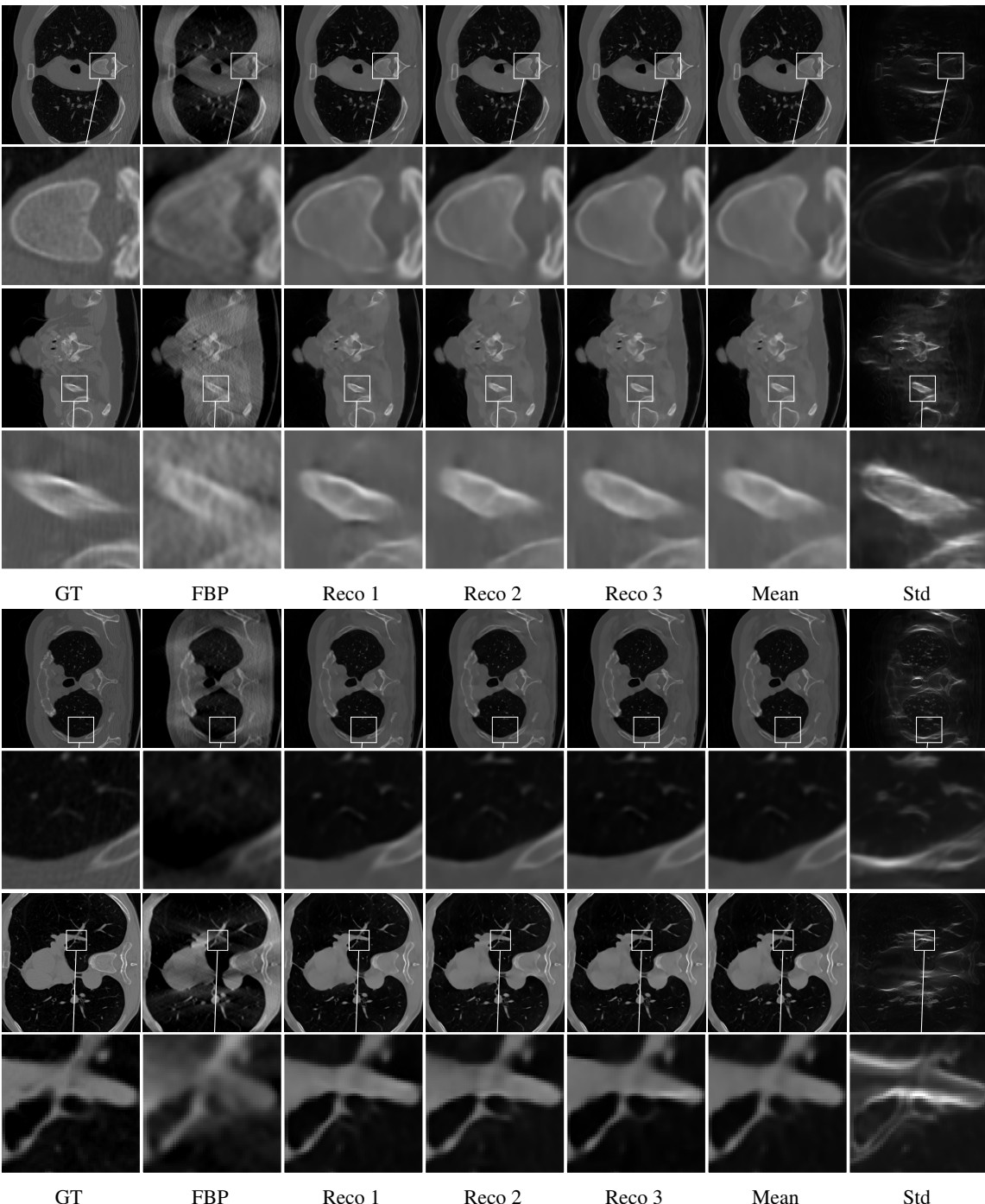

Figure 10: Additional generated posterior samples, mean image and pixel-wise standard deviation for limited angle computed tomography using conditional MMD flows.

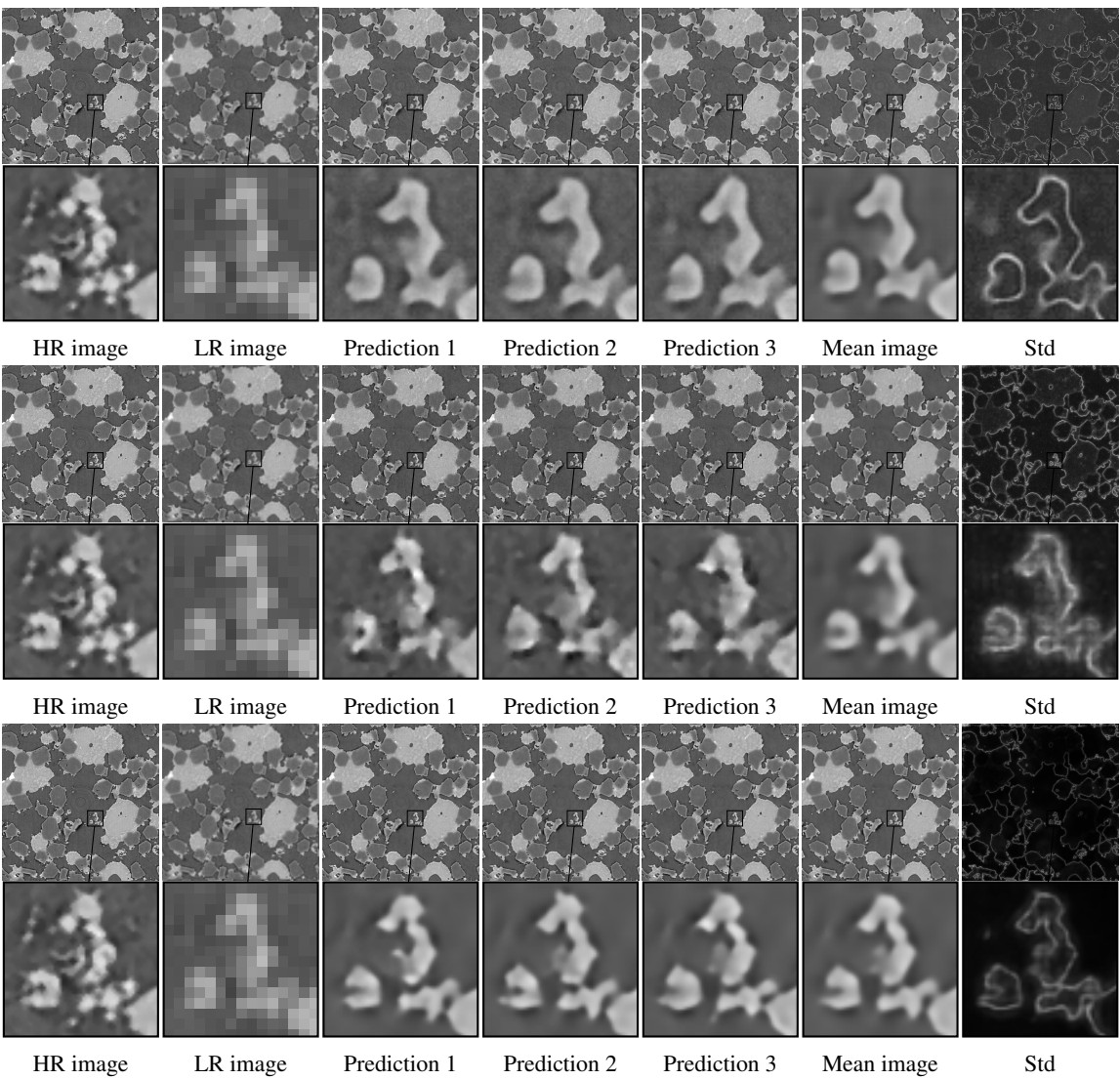

Figure 11: Different WPPFlow (top), SRFlow (middle) and conditional MMD flow (bottom) reconstructions of the HR image and their mean and pixelwise standard deviation (normalized). The zoomed-in part is marked with a black box.

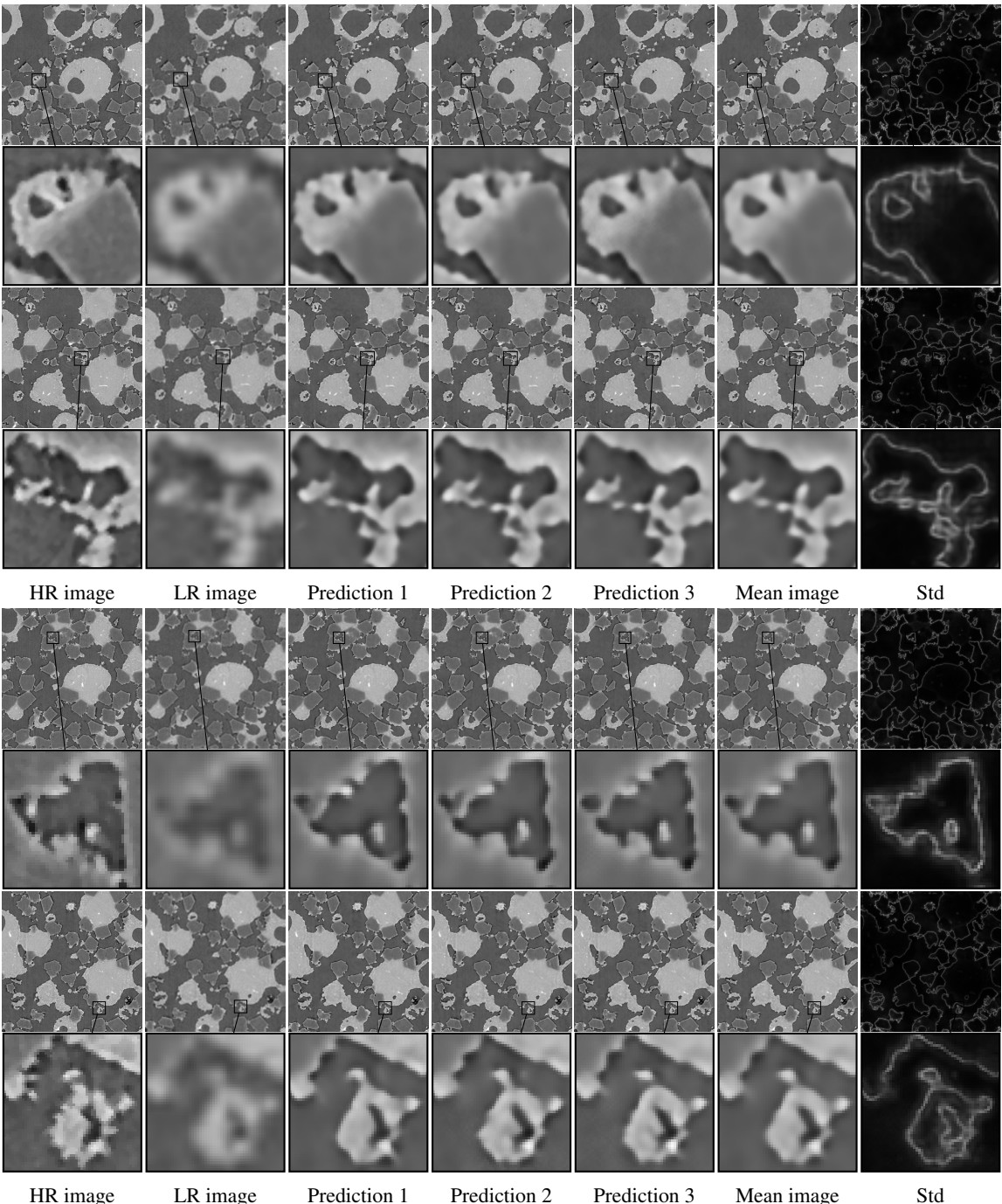

Figure 12: Additional conditional MMD flow reconstructions of the HR image and their mean and pixelwise standard deviation (normalized). The zoomed-in part is marked with a black box.

