# OpenReview forum: "Posterior Sampling Based on Gradient Flows of the MMD with Negative Distance Kernel"
_ICLR.cc/2024/Conference — ICLR 2024 poster_

### Official Review · Reviewer_MCWB · 2023-10-20

**Soundness:** 3 good
**Presentation:** 3 good
**Contribution:** 3 good
**Rating:** 6
**Confidence:** 4

**Summary:**

The paper introduces conditional flows of the Maximum Mean Discrepancy (MMD) with the negative distance kernel for posterior sampling and conditional generative modeling. The joint distribution of the ground truth and the observations is approximated using discrete Wasserstein gradient flows, and an error bound for the posterior distributions is established. it is proven in the paper that the particle flow within our method indeed functions as a Wasserstein gradient flow of an appropriate functional. The paper's efficacy is demonstrated through various numerical examples, encompassing applications such as conditional image generation and the resolution of inverse problems, including superresolution, inpainting, and computed tomography in low-dose and limited-angle scenarios.

**Strengths:**

* The proposal of MMD flows with a "generalized" kernel kernel which is also known as energy distance or Cramer distance is new.
* The paper can prove that the particle flow with the generalized MMD is indeed a Wasserstein gradient flow of an appropriate function.
* The paper uses the MMD flows in the setting of sampling from the posterior which is interesting and new.
* Experiments are conducted on class-conditional image-generation (MNIST, FashionMNIST, and CIFAR10) and inverse problems with medical images.

**Weaknesses:**

* There is no quantitative comparison in class-conditional image-generation with previous works e.g., score-based generative modeling (without using labels). Similarly, score-based generative models can also be used in medical image inverse-problem [1].
* There is no comparison with Sliced Wasserstein Gradient flows e.g., with JKO scheme. [2]
* Considering discrete flows is quite restricted.

[1] Solving Inverse Problems in Medical Imaging with Score-Based Generative Models.
[2] Efficient Gradient Flows in Sliced-Wasserstein Space

**Questions:**

* Standard Sliced Wasserstein is not optimal, there are other variants e.g., [3],[4]. Is standard SW preferred in this setting?
* Can the proposed MMD flows be seen as a debiased version of Sliced Wasserstein gradient flow in the setting of discrete flows?

[3] Generalized Sliced Wasserstein Distances
[4] Energy-Based Sliced Wasserstein Distance

---

> ### Author Response · Authors · 2023-11-16
> **Rebuttal**
>
> Thank you very much for the review. To address the first two points from the "weaknesses" part, we added a comparison for class-conditional image generation with the previous method of (Du et al. 2023) which is based on sliced Wasserstein gradient flows. For details, we refer to the general answer.
> We address your other comments below.
>
>
> - Regarding other slicing variants:
> Consider other slicing procedures could be an interesting line of future research.
> In this paper, we focused on the standard mean-slicing procedure, because of its theoretical tractability.
> We added a sentence to the future-work section that we want to generalize our results for other slicing variants and transfer these results to (sliced) MMD functionals.
> Note that the focus of the paper is on conditional Wasserstein gradient flows of the MMD functional.
> Clearly, we have for our special negative distance kernel that "MMD = sliced MMD" which can be used in the numerical part,
> but plays no role in the rest of the paper.
>
> - Regarding gradient biases: (Bellemare et al. 2017) show that Wasserstein distances have biased sample gradients. Since in 1D Wasserstein and sliced Wasserstein coincide, this probably also holds true for the sliced Wasserstein distance, even though we are not aware of a formal proof of this statement.
> In Appendix A and Proposition 3 of the same paper (Bellemare et al. 2017) it is shown that the energy distance (= MMD with negative distance kernel) has unbiased sample gradients.
> In 1D, the energy distance coincides with the so-called Cramer distance, which is presented in Bellemare et al. 2017) as "debiased Wasserstein distance". In this sense, one could indeed view our MMD flows as "debiased sliced Wasserstein gradient flows". However, since we do not focus on gradient biases, we do not want to use this term within the paper.
>
> (Bellemare et al, 2017) The Cramer Distance as a Solution to Biased Wasserstein Gradients

---

> > ### Comment · Reviewer_MCWB · 2023-11-21
> > **Response to authors**
> >
> > Thank you for your replies,
> >
> > My concerns are partially addressed i.e., a baseline was added. I raised my score to 6.
> >
> > Best regards,

---

### Official Review · Reviewer_mNmx · 2023-10-28

**Soundness:** 2 fair
**Presentation:** 3 good
**Contribution:** 2 fair
**Rating:** 5
**Confidence:** 4

**Summary:**

This paper proposes conditional MMD flows with the negative distance kernel for posterior sampling and conditional generative modelling. By controlling the MMD of the conditional distribution using the MMD of the joint distribution, the paper provides a pointwise convergence result. In addition, the paper shows that the proposed particle flow is a Wasserstein gradient flow of a modified MMD functional, and hence provides some theoretical guarantee for [1]. Finally, the paper experiments on several image generation problems and compares with other conditional flow methods.

[1] C. Du, T. Li, T. Pang, S. Yan, and M. Lin. Nonparametric generative modeling with conditional slicedWasserstein flows. In A. Krause, E. Brunskill, K. Cho, B. Engelhardt, S. Sabato, and J. Scarlett (eds.), Proceedings of the ICML ’23, pp. 8565–8584. PMLR, 2023.

**Strengths:**

1. The paper is well-written and clearly-organized.
2. The paper proves that the proposed particle flow is a Wasserstein gradient flow of an appropriate functional, thus providing a theoretical justification for the empirical method presented by [1].
3. Abundant generated image samples are shown in the experiments.

[1] C. Du, T. Li, T. Pang, S. Yan, and M. Lin. Nonparametric generative modeling with conditional slicedWasserstein flows. In A. Krause, E. Brunskill, K. Cho, B. Engelhardt, S. Sabato, and J. Scarlett (eds.), Proceedings of the ICML ’23, pp. 8565–8584. PMLR, 2023.

**Weaknesses:**

1. The novelty of the proposed method appears to be limited, since it is mainly the Generative Sliced MMD Flow [1] method applied to conditional generative modelling problems. Additionally, the proof of Theorem 3 partially follows [2].
2. The theoretical comparison with different kernels (Gaussian, Inverse Multiquadric and Laplacian [1]) and discrepancies (KL divergence, W_1 [2] and W_2 [3] distance) in Theorem 2 is insufficient.
3. The numerical results of image generation lack comparison with other methods like Generative Sliced MMD Flow in [1]. It would be better to compare the FID scores for different datasets and various methods like [1], since the proposed method adopts the computational scheme of Generative Sliced MMD Flow. It would be beneficial to compare with Conditional Normalizing Flow in the superresolution experiment and with WPPFlow, SRFlow in the computed tomography experiment.


[1] J. Hertrich, C. Wald, F. Altekrüger, and P. Hagemann. Generative sliced MMD flows with Riesz kernels. arXiv preprint 2305.11463, 2023c

[2] F. Altekrüger, P. Hagemann, and G. Steidl. Conditional generative models are provably robust: pointwise guarantees for Bayesian inverse problems. Transactions on Machine Learning Research, 2023b.

[3] F. Altekrüger and J. Hertrich. WPPNets and WPPFlows: the power of Wasserstein patch priors for superresolution. SIAM Journal on Imaging Sciences, 16(3):1033–1067, 2023.

**Questions:**

1. The paper states that MMD combining with the negative distance kernel results in many additional desirable properties, however it lacks convergence rate or discretization error analysis because “the general analysis of these flows is theoretically challenging”. Regarding this problem, what is the advantage of MMD over other discrepancies like Kullback–Leibler divergence or the Wasserstein distance especially for conditional generative modelling problems?
2. Is it possible to provide a discretization error analysis between discrete MMD flow and the original continuous MMD flow?

---

> ### Author Response · Authors · 2023-11-16
> **Rebuttal**
>
> Many thanks for your comments. We added several comparisons in the numerical part. In particular, we added a comparison with WPPFLows on CT and a FID comparison for class-conditional image generation.
>
> ## Regarding the novelty
>
> Note this is **not only** "a generative Sliced MMD Flow [1] method applied to conditional generative modelling problems".
> Our paper is about **conditional Wasserstein gradient flows of the MMD functional**.
> There is just a small hint that "MMD = sliced MMD" for the negative distance kernel, see [1], which is irrelevant for the theory part.
>
> The consideration of the conditional flows brings indeed a lot of new theoretical challenges.
> Two of them are addressed in our paper:
>
> i) Conditional generative models approximate the joint distribution  by learning a mapping $T$ such that
> $P_{X,Y} \approx P_{T(Z,Y)),Y}$,
> but in fact we are interested in the posterior distributions $P_{X|Y=y}$.
> In this paper, we prove error bounds between posterior and joint distributions within the MMD metric.
> To this end,
> we use relations between measure spaces and RKHS as well as Lipschitz stability results
> under pushforward measures which are quite involved.
> These results are new and highly non-trivial to prove. Clearly they are based on preliminary work
> in particular on [2], but we just cited the necessary results and use them in completely new proofs.
>
> ii) We want to characterize a **conditional** Wasserstein gradient flow of a functional $F$ as
> **usual** Wasserstein gradient flow of a modified functional to better understand the behaviour of the first one.
> To establish a relation to Wasserstein gradient flows, we locally lift the $\mathbb R^{N d}$ into the Wasserstein space using local isometries. As a byproduct we give a theoretical explanation for the results of (Du et al. 2023).
>
> We have rewritten the contributions paragraph in the introduction to highlight the points i) and ii).
>
> In the applications part, the conditional MMD flows for large-scale imaging inverse problems like computed tomography is new.
> Of course we use [1] for reducing the computational cost and we agree that this is crucial to obtain a tractable algorithm, but this is not the point of our contribution.
>
> ## Other comments
>
> - Thanks for the suggestion to compare with a conditional normalizing flow for superresolution and with SRFlow for CT. We would like to point out that SRFlow is the same as a conditional normalizing flow, where the authors adapted the architecture for superresolution. We added a comment about that in the paper.
>
> - Different kernels: Since the negative distance kernel is the only kernel, where we can compute the gradient of MMD in $O(N\log(N))$ instead of $O(N^2)$, it is the only kernel, where our method is applicable. From a theoretical viewpoint, generalizing Theorem 2 to a larger class of kernels would be nice and is a current line of our research. For the KL divergence, we have to restrict to absolutely continuous measures, but get a stronger version of Theorem 2 with an equality statement. We refer to Remark 7 in Appendix A.2 for a detailed discussion of Theorem 2 with KL or Wasserstein-1.
>
> - Advantage over KL and Wasserstein: Neither the KL divergence nor the Wasserstein distance admit a closed-form two-sample formulation. For the KL divergence, we have to estimate the density (or the score) of one of the measures,
> for Wasserstein we have to solve an optimization problem.
> In contrast, we can just compute the MMD and its derivative directly on two sets of samples.
> In order to lower the complexity of the computations, one can apply the slicing procedure from [1]
> which admits a provable error bound.
>
> - Regarding error bounds between discrete and continuous MMD flows: In the one-dimensional case, (Carillo et al. 2020) proved that the discrete MMD flows with negative distance kernel converge in the mean-field limit to the continuous ones and we cited this in our paper.
> In higher dimensions, the MMD functional with negative distance kernel is not $\lambda$-convex along generalized geodesics such that most of the theory of Wasserstein gradient flows is no longer applicable. In (Hertrich et al. 2022), the authors derive analytic formulas for MMD flows with negative distance kernel in higher dimensions.
> Despite these examples, we are not aware of any theoretical results regarding existence, uniqueness or convergence results for such flows for $d\geq 2$.
> Therefore, we feel that there is a long line of theoretical work which has to be done before a proof of such error bounds would be reachable. Therefore they are beyond the scope of our paper.
>
> ## References:
>
> [1], [2] from the review
>
> (Carillo et al. 2020) Measure solutions to a system of continuity equations driven by newtonian nonlocal interactions.
>
> (Hertrich et al. 2022) Wasserstein steepest descent flows of discrepancies with Riesz kernels.
>
> (Hertrich et al. 2023) Generative sliced MMD flows with Riesz kernels.

---

### Official Review · Reviewer_eQGc · 2023-10-30

**Soundness:** 3 good
**Presentation:** 3 good
**Contribution:** 3 good
**Rating:** 8
**Confidence:** 3

**Summary:**

In this paper, conditional MMD flow with negative distance kernel is introduced.
The model's stability is proven by bounding the expected approximation error of the posterior distribution.

Through theoretical justification, the authors obtain convincing results by neglecting the velocity in the y-component in sliced Wasserstein gradient flows.
Then, the power of the method is also demonstrated by numerical examples including conditional image generation and inverse problems.

**Strengths:**

1. The theoretical justification of the proposed method is clear and detailed.
2. Several experiments are conducted to prove the power of the method.
3. Introducing negative distance kernel to MMD is a good idea and contributions are well-described.

**Weaknesses:**

As mentioned by the authors, the proposed approach has some limitations:

1. The model is sensitive to forward operator and noise type.
2. Lack of meaningful quality metrics to evaluate the results.
3. Realism of the computed tomography experiment results can not be guaranteed.

**Questions:**

1. Except computed tomography experiment, only visulization results of other experiments are given in the paper, however, it is difficult to quantitatively evaluate the result and to compare with other method. Hence, evaluation metrics need to be introduced or self-defined.

2. The related work: Neural Wasserstein gradient flows for maximum mean discrepancies with Riesz kernels, proposed similar method, what is the strength and advantage over it? and what about the performance difference?

3. Why chosing UNet? Is there a significant difference in the effect of choosing other models such ResNet and transformer.

4. As Fig.7c shows, inpainting results of CIFAR are not good enough, the generated images differ from each other greatly at the unobserved part, what is the reason? and are there any solutions to improve it.

---

> ### Author Response · Authors · 2023-11-16
> **Rebuttal**
>
> Thank you very much for your comments. We added quantitative evaluations of our method as outlined in the general answer. We address your other comments below.
>
> - Regarding the sensitivity to the forward operator/noise: We want to clarify that this issue is not specific for our method. Instead, any supervised learning method for inverse problems is sensitive to mismatches between the training and testing models.
> We emphasize this issue in the limitations section, because it is particularly relevant for medical imaging applications, where such mismatches are unavoidable. However, in our paper we want to demonstrate the usability of our method for highly ill-posed and high-dimensional imaging inverse problems. In particular, calibrating it for clinical applications is not within the scope of our paper.
> This is common-sense in the machine learning community and unfortunately most authors do not even mention that.
>
> - The paper (Altekrüger et al, 2023) **does not provide any "proper" generative model**.
> Instead the authors consider forward and backward schemes for non-discretized Wasserstein gradient flows. Consequently, they have to train a generative model **in each step of the gradient flow**. This allows an extensive analysis of such flows, but is computational costly and therefore not realizable for large problems. Indeed, their MNIST example does not consider the whole dataset but only a subset of a few hundred images.
>
> - The U-Net is a standard choice for many imaging applications as well as in diffusion models, see (Huang et al, 2021).
> We did also some brief experiments with a lightweight transformer network and achieved slightly worse results.
> We think that there is not a very large difference as long as the considered network is expressive enough.
> - Regarding the inpainting example: CIFAR10 is a highly diverse dataset such that a large variation of the generated samples for inpainting is natural. Consequently, the generated images have to differ from each other greatly. Note that for less diverse datasets like (Fashion)MNIST or CelebA, we obtain much less variation in the reconstructed images.
>
>
> References
>
> (Du et al, 2023) Nonparametric Generative Modeling with Conditional Sliced-Wasserstein Flows, Du, Li, Pang, Shuicheng, Lin, ICML 2023
>
> (Altekrüger, 2023) Neural Wasserstein Gradient Flows for Discrepancies with Riesz Kernels, Altekrüger, Hertrich, Steidl, ICML 2023
>
> (Huang et al, 2021): A variational perspective on diffusion-based generative models and score matching, Huang, Lim and Courville, NeurIPS 2021

---

### Official Review · Reviewer_PLVu · 2023-10-31

**Soundness:** 3 good
**Presentation:** 3 good
**Contribution:** 2 fair
**Rating:** 5
**Confidence:** 2

**Summary:**

This paper proposes a conditional flow of the MMD with the negative distance kernel, which can be further implemented by conditional generative neural networks with application in image generation, inpainting, and super-resolution. The authors derive the convergence of the posterior under some certain stability conditions, and relate it to a Wasserstain gradient flow. Those results extend previous investigation for sliced Wasserstein flow. The work is relatively theoretical and lacks a thorough comparison with other generative models.

**Strengths:**

The paper presents some interesting theories, and extends the analysis on sliced Wasserstein gradient flow.

**Weaknesses:**

1. It would be better to elaborate on the pros and cons of using a negative distance kernel (efficiency, sample complexity, etc).

2. The contribution is not entirely clear. Could the author comment on the effectiveness/efficiency/novelty/difficulty of the proposed method?

3. A highlight of the proof techniques used by the authors to address gradient flows with respect to MMD with negative distance kernel without mean-field approximation would help to improve the importance of this work.

**Questions:**

1. In Equation 4, $T$ is defined, however $T_\sharp$ is not defined.


2. Is it possible to validate the error bound via numerical experiments somehow?


3. Could the author comment on the difference between the proposed analysis and sliced Wasserstein flow, as the implementation is still based on the sliced version of it?

---

> ### Author Response · Authors · 2023-11-16
> **Rebuttal**
>
> Thank you very much for the review. We added comparisons with other generative models as outlined in the general answer. We address your other comments below.
>
> ## Main proof techniques and contributions
>
> We have two main theoretical contributions in addition to numerical applications:
>
> i) Conditional generative models approximate the joint distribution by learning a mapping such that
> $P_{X,Y} \approx P_{T(Z,Y)),Y}$,
> but in fact we are interested in the posterior distributions $P_{X|Y=y}$.
> In this paper, we prove error bounds between posterior and joint distributions within the MMD metric.
> To this end,
> we use relations between measure spaces and RKHS as well as Lipschitz stability results
> under pushforward measures which are quite involved.
>
> ii) We want to characterize a **conditional** Wasserstein gradient flow of a functional $F$ as
> **usual** Wasserstein gradient flow of a modified functional to better understand the behaviour of the first one.
> To establish a relation to Wasserstein gradient flows, we locally embed the $\mathbb R^{N d}$ into the Wasserstein space using local isometries. As a byproduct we give a theoretical explanation for the results of (Du et al. 2023).
>
> iii) Numerically, we approximate our conditional MMD flows by conditional generative neural networks and apply them in various settings like class conditional image generation, image restoration and CT.
>
> We have rewritten the contributions paragraph in the introduction to highlight the points i), ii) and iii).
>
> ## Difference to sliced Wassestein flows
>
> We want to emphasize that our proposed method **does not use sliced distances**, also not the sliced Wasserstein distance.
> We consider conditional gradient flows for the functional $F$=MMD within the $W_2$ geometry.
> In contrast, conditional sliced Wasserstein gradient flows (Du et al. 2023) consider gradient flows for the functional $F$=sliced Wasserstein within the $W_2$ geometry. Consequently, both methods act in the $W_2$ geometry, but minimize **completely different functionals**.
> Our paper contains just a  hint that for the negative distance kernel (and only for this kernel) it holds
> ``MMD = sliced MMD''.
> This is an exceptional nice property proven in (Hertrich et al. 2023) which we clearly
> exploit in the numerical part for speeding up the computations.
>
> ## Other comments:
>
> - Regarding the numerical verification of bounds we have to note that this is a worst case bound.
> Therefore it is hard to do numerical verification (as obtaining the ground truth posterior is hard/impossible in high dimensions)
> and it is likely that real world examples actually attain better rates.
>
> - We added some nice properties of the negative distance MMD (efficient calculation and good sample complexity) in the introduction.
>
> - $T$#$\mu$: This denotes the pushforward measure of $\mu$. We included the definition in the paper.
>
> ## References
>
> (Hertrich et al. 2023) J. Hertrich, C. Wald, F. Altekrüger, and P. Hagemann. Generative sliced MMD flows with Riesz kernels. arXiv preprint 2305.11463

---

### Author Response · Authors · 2023-11-16
**General answer**

We would like to thank all reviewers for the evaluation of our paper.
We are glad that the reviewers found the theoretical contributions well-justified (reviewer eQGc) and interesting (reviewer PLVu), well-written and presented well (all reviewers) and with good novelty (reviewer MCWB).
Based on the reviewers comments, we carefully updated the paper. Changes are indicated in blue.
In particular, we **extended the quantitative comparisons**:

- class conditional image generation: we added a comparison with (Du et al. 2023) which is (to the best of our knowledge) so far the only other paper on conditional Wasserstein gradient flows. We evaluate the class conditional FID for the conditional image generation examples.
Here we compute the FID between samples from the conditional generative model applied on for fixed class and all test samples corresponding to the same class.
We observe that we significantly outperform the method from (Du et al. 2023) on MNIST and CIFAR10 and are on par (slightly worse) on FashionMNIST, see Fig. 2.

- for inverse problems: since for each observation, there is only one ground-truth available, we cannot evaluate FIDs in this case. As the best quality metric, which we can access, we measured the average PSNR/SSIM from the generated images to the ground truth. We compare with conditional normalizing flows/SRFlow and WPPFlow. Also for these experiments, we outperform the comparisons. We added a comparison with WPPFlow on CT.

Regarding the other questions and comment we answer to each review separately.

---

### Meta-Review · Area_Chair_9FW1 · 2023-12-09

**Metareview:**

The paper proposes conditional MMD flows with the negative distance kernel for posterior sampling and conditional generative modeling. The reviewers believed the paper was well argued and derived. The connection to Wasserstein gradient flows was thought to be good. Some concern was raised about comparisons, but the net sentiment was that the paper should be accepted.

**Justification For Why Not Higher Score:**

Not enough enthusiasm for paper

**Justification For Why Not Lower Score:**

Reviewers seemed to engage with the idea better than other papers.

---

### Decision · Program_Chairs · 2024-01-16

Accept (poster)